# A Survey of SAR Image Target Detection Based on Convolutional Neural Networks

**Ying Zhang** [1,*,†] [iD] **and Yisheng Hao** [2,†]

1   College of Information Engineering, Shanghai Maritime University, Shanghai 201306, China
2   Institute of Logistics Science and Engineering, Shanghai Maritime University, Shanghai 201306, China
*   Correspondence: yingzhang@shmtu.edu.cn; Tel.: +86-2138-282-864
†   These authors contributed equally to this work.

**Abstract:** Synthetic Aperture Radar (SAR) target detection is a significant research direction in radar information processing. Aiming at the poor robustness and low detection accuracy of traditional detection algorithms, SAR image target detection based on the Convolutional Neural Network (CNN) is reviewed in this paper. Firstly, the traditional SAR image target detection algorithms are briefly discussed, and their limitations are pointed out. Secondly, the CNN's network principle, basic structure, and development process in computer vision are introduced. Next, the SAR target detection based on CNN is emphatically analyzed, including some common data sets and image processing methods for SAR target detection. The research status of SAR image target detection based on CNN is summarized and compared in detail with traditional algorithms. Afterward, the challenges of SAR image target detection are discussed and future research is proposed. Finally, the whole article is summarized. By summarizing and analyzing prior research work, this paper is helpful for subsequent researchers to quickly recognize the current development status and identify the connections between various detection algorithms. Beyond that, this paper summarizes the problems and challenges confronting researchers in the future, and also points out the specific content of future research, which has certain guiding significance for promoting the progress of SAR image target detection.

**Keywords:** object detection; synthetic aperture radar (SAR); convolutional neural network (CNN)

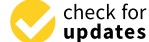



## 1. Introduction

Radar is a tool that detects object distance, radial velocity as well as height by transmitting electromagnetic waves. SAR is a characteristic representative of existing radar equipment [1]. Compared with conventional optical sensors, SAR is not confined by illumination intensity, weather, or other factors in imaging or detection, and has other advantages, such as all-weather and long distance [2]. In contemporary years, with the continuous maturity of SAR imaging techniques and the continuous improvement of image resolution, SAR image post-processing has become a hot issue.

As a part of the SAR image post-processing, SAR target detection intends to rapidly and efficiently extract the orientation and position of the target from some complicated scenes. SAR target detection is a crucial part of SAR target automatic recognition, too. The detection precision determines a series of subsequent projects.

In recent years, deep learning (DL) [3] has evolved rapidly at an alarming rate and has achieved good application results in numerous fields. DL adaptively extracts data features and learns data distribution by constructing a deep neural network. When processing tasks, DL can completely mine the target feature information. In the process of repeated training, the DL algorithm can learn the deeper, higher dimensional, and more comprehensive image's features, and then find the optimal solution by gradient descent method to detect the target. As a mainstream algorithm in the domain of computer vision

(CV), CNN can effectively extract low-dimensional features and high-dimensional features of images due to its hierarchical structure. It is widely used in target detection [4], semantic segmentation [5], and other fields. Inspired by this, CNN has also received increasing attention from researchers in SAR image target detection. Due to the SAR-specific imaging mechanism, there will be a lot of clutter and noise in SAR images, and the signal-to-noise ratio is frequently moderately weak, which causes considerable difficulties in the feature extraction of SAR images. CNN has a deep feature extraction structure and powerful feature extraction capability. Hence, it has a natural advantage when facing SAR image processing tasks.

So far, the state-of-the-art CNN-based target detection algorithms can be approximately divided into two categories: (1) a two-stage target detection algorithm based on candidate boxes; (2) a one-stage target detection algorithm based on regression [6]. Different from traditional target detection algorithms based on a sliding window, a target detection algorithm based on CNN can learn image features in the data, so it can avoid manual feature selection and reduce human error. Therefore, the method possesses good detection accuracy and robustness.

The paper's dominating contributions can be summarized as follows:

1.  For the traditional SAR image target detection algorithm, we divided the traditional detection algorithm into three categories, and studied the detection algorithm of each category with the relevant references, analyzed the basic idea, advantages, and disadvantages of different algorithms under the same category. Based on this, we summarized the characteristics of these three algorithms, which then lead to the necessity of using CNN for SAR image target detection.
2.  We analyzed the fundamental theory and network structure of CNN and studied the SAR target detection data sets which are frequently used at present.
3.  Based on a mass number of references, we studied the CNN-based SAR image target detection. According to the main problems faced by CNN in SAR image target detection, we divided the literature review analysis into five categories. We summarized the innovative ideas of various improved algorithms. Simultaneously, we compared CNN with traditional SAR target detection algorithms and obtained the characteristics of various algorithms.
4.  The difficulties and challenges in the field of SAR image target detection were derived from the analysis of references, which pointed out the direction for future research.

The paper's full framework is arranged as follows. Section 1 is an introduction. Section 2 is a brief introduction to the research methodology. Section 3 introduces the SAR image target detection based on traditional algorithms. Section 4 analyzes the CNN's principle and introduces the progress of CNN in the RGB image field briefly. Section 5 introduces SAR image-related knowledge, including SAR image data sets and image preprocessing. In addition, Section 5 particularly analyzes the SAR image target detection algorithms based on CNN. Based on the above discussion, Section 6 proposes future research directions and analyzes some challenges. Finally, some conclusions of this paper are made in Section 7.

## 2. Research Methodology

This article is a review article, so the selection of references is extremely important. Based on reference [7] and combined with the paper's main idea, we have summarized the research methodology of this paper. From the point of the reference publication time, most of the articles are selected within the past five years, and a few articles are predominantly published between 2010 and 2015. This paper does not select literature with a longstanding gap, mainly to analyze the current research trends and summarize the latest scientific research results in recent years. In terms of selecting article journal databases, we primarily searched for relevant literature from MDPI, IEEE Xplore, Science direct, and other databases. For the research content, we review the SAR image target detection from two major aspects. One is the traditional algorithm, and the other is the CNN-based algorithm. Then these

two large aspects are subdivided into different small aspects. For the traditional algorithms, we divide the traditional algorithms into three categories. For CNN-based algorithms, we analyze from the perspective of CNN solving different SAR detection tasks. The purpose of doing this is to make our objectives clearer, the articles more representative, and the summary more comprehensive.

## 3. SAR Image Target Detection Based on Traditional Algorithm

The goal of SAR image target detection is to accurately and quickly extract the target in the image under a complex background. It can identify the target's category and accurately locate the target position. SAR image target detection usually consists of five stages. Here, the proposal detection and identifying the target are what we know as the detection stage. Each stage plays a different role. Figure 1 shows the specific process of SAR image target detection.

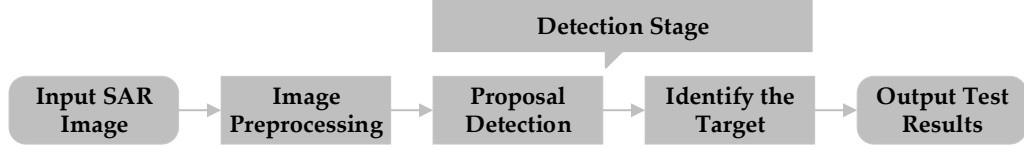

**Figure 1.** The flow diagram of SAR image target detection.

At present, the SAR image target detection method based on traditional algorithms is principally classified into three categories: detection algorithm based on structural feature (SF), detection algorithm based on gray feature (GF), and detection algorithm based on image texture feature (ITF). Specifically, SF comprises two types of algorithms, which are the target geometric feature extraction algorithm (TGFEA) and the target azimuth estimation algorithm (TAEA). GF contains a constant false alarm rate algorithm (CFARA). ITF contains a target fractal feature detection algorithm (TFFDA). Figure 2 shows the relationship between these algorithms. Next, some SAR image target detection based on traditional algorithms will be analyzed and discussed on the basis of some of the literature. Accordingly, we study the characteristics of each type of algorithm by comparison.

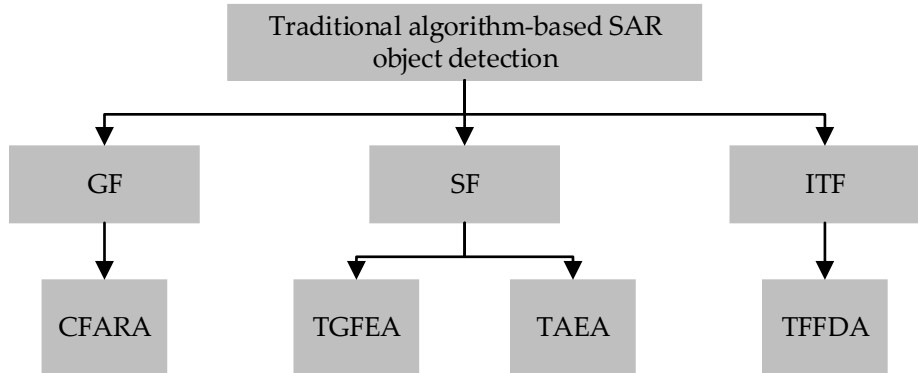

**Figure 2.** Classification of SAR image target detection based on traditional algorithms.

### 3.1. Object Detection Algorithm Based on Structural Features

The structural feature of a target is a considerably significant feature for target detection. Accurate extraction of structural features can provide a large amount of prior information for the algorithm. In general, the target structural characteristics incorporate the structure distribution, the target contour, and the target part's shape. Therefore, accurately extracting structural features is crucial for promoting algorithmic recognition speed and accuracy. Table 1 gives a brief overview of various improved algorithms mentioned in this section.

In [8], the author summarized many ship features in SAR images, including geometric features, transformation features, electromagnetic features, and local invariant features. Then they analyzed the physical meaning and calculation method of these features. Since the calculation method comes from optical remote sensing images, for SAR images, it is susceptible to the strong scattering target side lobe effect, which will have some effects on the precision of feature extraction. Reference [9] proposed a method for extracting the geometric structure features of ship targets. This method is based on azimuth estimation. For the calculation of azimuth, this method does not adopt the traditional regression analysis method and the minimum enclosing rectangle (MER) but adopts the moment estimation method. Through boundary approximation, this method can effectually eliminate some irregular shapes from the visual effect, consequently, enhancing the extraction accuracy of geometric features. Gu, D et al. [10] came up with a multi-feature joint algorithm to extract the size and azimuth of the target, using the "dichotomy" to precisely search the azimuth. Although the algorithm can effectively extract features, the complexity of the algorithm is high, which will have some negative effects on post-processing. In allusion to the defects of low precision and poor stability of traditional extraction algorithms, Reference [11] put forward a SAR image feature extraction algorithm based on fine segmentation. The algorithm uses Radon transform to effectually separate the object and background interference region. Then, the target area is subjected to threshold segmentation and morphological processing, which can efficaciously decrease the effect of side lobes. Then the elliptical shape constraint is used to segment the target area. Finally, by approximating the target area, the features can be efficaciously extracted. The simulation experiments demonstrate that this algorithm demonstrates high precision and strong robustness for the geometric feature extraction of marine ship targets.

**Table 1.** A brief overview of the target detection algorithms based on structural features.

| Algorithm Ideology | Advantage | Disadvantage |
| --- | --- | --- |
| Geometric feature extraction based on azimuth angle [9] | Higher accuracy and better stability | The process is complex and the application is limited |
| Multi-feature combination [10] | Effectively extracts the features | High complexity and unfavorable for subsequent processing |
| Geometric feature extraction based on fine segmentation [11] | More accurate segmentation and better performance | Features not yet combined with other methods to further exploit the algorithm |

### 3.2. Object Detection Algorithm Based on Gray Features

The most common detection algorithm based on gray features is constant false alarm rate (CFAR), which is a well-developed SAR target detection method. The basic principle of CFAR is as follows. Firstly, a certain false alarm rate threshold is set in advance. Secondly, the detection threshold is calculated according to the clutter characteristics in the background. Then, whether a pixel in the SAR image is the target pixel is determined. Finally, the target is detected [12]. The concrete process of CFAR is shown in Figure 3.

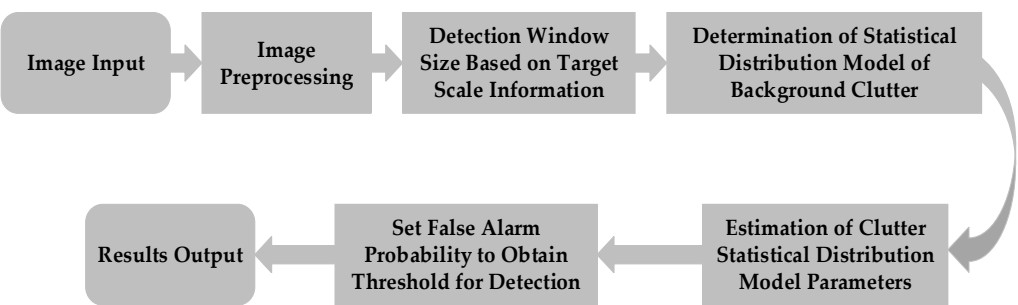

**Figure 3.** The flow chart of CFAR detection algorithm.

Since CFAR is a pixel-level detection algorithm, when detecting targets in SAR images, it is essential to evaluate the noise distribution. However, particularly in complex backgrounds, it is hard to attain a satisfactory detection effect through this single statistical distribution model. It frequently misses detection or provides false detection. Therefore, in some specific instances, it is necessary to improve the CFAR or combine some other algorithms to elevate the accuracy of SAR image target detection. Table 2 gives a brief overview of various improved algorithms cited in this section.

**Table 2.** A brief overview of the target detection algorithms based on gray features.

| Algorithm Ideology | Advantage | Disadvantage |
| --- | --- | --- |
| IB-CFAR in the complex environment [13] | Strong robustness and high detection rate | Partial false alarms would occur |
| Two-Parameter CFAR detection [14] | Low FAR and high precision | Worse robustness and universality, incomplete individual parameters |
| A CFAR algorithm based on shadow feature semantics [15] | Lower FAR | Large amount of computation, poor migration generalization ability |
| An improved CFAR based on similarity judgment and attention mechanism [16] | High efficiency, strong ability of the multi-scale target detection | High requirements for device memory, unfavorable detection of large numbers of images |
| CFAR based on adaptive background clutter model [17] | Lower missed detection rate and higher accuracy | Long detection time and lower efficiency |

Aiming at the deficiencies of the classical CFAR with its large errors, false alarms, and missed detections in complex environments, Ai, J et al. [13] proposed an improved bilateral CFAR ship detection algorithm (IB-CFAR). The algorithm mainly involves three stages: intensity level partition, intensity-space domain information fusion, and parameter estimation after clutter truncation. Firstly, a non-uniform quantization method is used to classify the image intensity, so that the target's similarity information is enhanced. Secondly, the adaptive intensity-space domain information fusion is used to considerably heighten the contrast information between the target and the ambient clutter. Finally, a clutter phase method with adaptive truncation depth is designed based on OR-CFAR. Experiments demonstrate that the algorithm has excellent detection performance in the multi-densely distributed weak target environment. Compared with the pre-existing algorithms, the detection rate is upgraded by 5%, and the false alarm rate (FAR) is decreased by 10%. The robustness of the algorithm is further significantly enhanced.

To lower the influence of strong target side lobe and speckle noise on detection performance, Chang, J [14] proposed an improved two-parameter constant false alarm rate (2P-CFAR) detection algorithm. A non-local mean filtering method is applied so as to better restrain speckle noise. At the same time, the minimum circumscribed rectangle of the candidate target is segmented horizontally and vertically, so that the aspect ratio of the obtained ship is more accurate. Experiments on GF-3 satellite SAR images show that the algorithm markedly reduces the influence of side lobe and speckle noise, and the FAR is also reduced, which verifies the effectiveness of the method. In [15], Huang Y put forward a novel CFAR algorithm based on semantic features to detect objects in high-resolution SAR images. Simulation experiments on the MiniSAR dataset prove that compared with traditional detection algorithms, the method can efficaciously reduce the false alarm target and reduce the FAR.

In allusion to the low performance of traditional CFAR for multi-scale target detection, Qu, Y [16] proposed an algorithm based on the object-likeness judgment by incorporating the human attention mechanism. The algorithm first uses the method of resemblance judgment to extract the candidate regions in the image and then obtains the window size of the CFAR background by the target's size. In the meantime, the integral image method is combined to optimize the CFAR. Through comparative experiments, the algorithm could greatly enhance the detection efficiency and has better multi-scale feature extraction ability.

In order to resolve the matter where the clutter model in the wide-swath SAR image is mismatched in the non-adaptive region, which generates the degradation of CFAR detection performance, Lin, X [17] proposed a CFAR algorithm based on an adaptive background clutter model. A highlight of the algorithm is that the target's clutter environment can be determined by the multi-scale statistical variance of the background window. The corresponding background clutter distribution model can be adaptively selected, that is, the log-normal distribution model is used in the uniform region, and the K distribution model is used in the non-uniform region. Through experimental comparison and analysis, it is concluded that the missed detection is significantly reduced, and the detection accuracy is greatly enhanced, which can be used for ship detection in board areas of the sea.

### 3.3. Object Detection Algorithm Based on Texture Features

In an image, the texture is a local image feature, which is formed by the arrangement of pixel values in a certain area according to specific rules. For SAR images, due to the particularity of their imaging characteristics, the texture features are strikingly different from those of ordinary optical RGB images. Therefore, it is quite essential to adopt a detection algorithm different from optical images when detecting targets in SAR images. In general, texture features could be split into four categories:

- Statistical texture features;
- Model texture features;
- Structural texture features;
- Signal-processing texture features [18].

Table 3 gives a brief overview of various improved algorithms cited in this section.

**Table 3.** A brief overview of the target detection algorithms based on image texture features.

| Algorithm Ideology | Advantage | Disadvantage |
|---|---|---|
| Target detection by using improved fractal feature [19] | Low FAR, better spatial resolution and more accurate position indication | As the background complexity increases, the FAR also increases |
| Algorithm of combining single-scale and multi-scale features [20,21] | Lower FAR, strong ability to distinguish targets, high accuracy | Imperfection classification techniques, incomplete comparison |
| Mean extended fractal [22] | Better resolution for bright and dark targets | Poor algorithm robustness |

At present, fractal features are broadly applied in SAR image target detection. In [19], Cheng gave a detection method, which was based on improved fractal features and extended fractal features in single and complex backgrounds. The simulation results illustrate that the improved fractal feature has a lower false alarm rate and more accurate positioning in single and complex backgrounds. In References [20,21], Charalampidis, D presented the method of wavelet fractal (WF) feature. This method uses a rotation invariant feature set for image texture segmentation as well as classification, so that the scale-related texture features are well characterized, and then achieved satisfactory results. Based on the change rate of the image gray mean, Reference [22] proposed the target detection algorithm of mean expansion fractal. The algorithm applies the mean expansion fractal to calculate the image's mean change and then realizes the target detection. Compared with the traditional extended fractal detection algorithm, the raised algorithm has better bright-dark and adjacent target discrimination ability as well as stronger target resolution.

### 3.4. Chapter Summary

Based on the previous literature research, it can be seen that nowadays, for SAR image target detection, considerable progress has been made by using traditional algorithms to detect targets. In a sense, these improvements to traditional algorithms have promoted the progress of SAR image target detection. However, after analyzing the above literature,

we also know that, based on traditional methods, there are still many shortcomings. The uppermost major problems are low detection accuracy, high missed detection rate or false detection rate, and poor algorithmic robustness. Specifically, the target detection algorithm based on structural features is easily affected by background clutter and requires some part of prior information. These deficiencies marginally impact the algorithmic performance. For the target detection algorithm based on gray features, although the CFAR algorithm is easier to implement than other algorithms and its computational complexity is small, this algorithm is a pixel-level detection algorithm. Hence it will inevitably ignore some target structural information, thus its generalization performance and robustness are poor. In the meantime, the CFAR algorithm heavily relies on the statistical distribution of the background clutter, so the target detection performance is bad in complex large scenes. For the detection algorithm based on image texture features, because the algorithm needs to extract texture features manually, it often demands a lot of manpower and material resources in the design process, which is relatively time-consuming, so algorithmic timeliness is not satisfactory. Therefore, a powerful algorithm is still essential for accurate and efficient target detection in SAR images.

In recent years, the most popular field is artificial intelligence (AI). The rapid development of AI has benefited from the application of deep learning algorithms. Deep learning is a kind of algorithm that constructs a deep network to discover the distributed feature representation of the data. The continuous process of deep learning provides new thinking and methods for target detection. Compared with traditional algorithms, the outstanding advantages of deep learning for target detection can be summarized as follows.

1. The model has high classification accuracy. Deep learning can fully make use of deep networks and use nonlinear activation functions to conduct layer-by-layer nonlinear transformation, which has a better approximation effect on complex functions.
2. Deep learning can accurately extract high-level features and avoid complex manual feature extraction, which greatly reduces the workload.
3. When the amount of data required for the model is substantial enough, the robustness and generalization of the algorithm will be relatively strong, and it has favorable adaptability to some complex environments as well.

CNN is a typical deep-learning model. It uses convolution operations and nonlinear mapping to effectively extract target features. It is a deep learning model extensively used in the field of computer vision (CV).

Based on the above analysis and discussion, there exists an obligation to further study CNN and introduce the advantages of CNN into SAR image target detection, so as to inject new vitality into this field.

## 4. Convolutional Neural Network

### 4.1. Basic Theory of CNN

In today's image field, CNN is a widely used network model. The obvious distinction between this network model and the general neural network is whether it contains convolution operation. In CNN, the role of convolution operation is feature extraction, which lays the foundation for the next image processing task. There are three momentous ideas in CNN, which provide thinking for scholars to continuously improve the convolutional neural networks at different levels. These three ideas are the locally connected layer, weight sharing, and sampling layer [23]. These operations can improve the network's performance and lower the risk of network overfitting.

#### 4.1.1. Locally Connected Layer

Unlike fully connected neural networks, CNN utilizes a local connection. If each pixel in the image is regarded as a neuron, then each output neuron of the fully connected network links all the neurons in the image while CNN only links a small number of adjacent neurons in space. Figure 4 shows the specific architecture. There are two reasons for CNN to adopt local connections. Firstly, for images, local pixels are closely associated, and the

correlation between pixels at farther distances is weaker. Therefore, each neuron does not need to perceive the total image but solely needs to perceive the local area. Then, the local information of low-level perception is synthesized at the high level to obtain global information. The second reason is to decrease the number of network parameters and decrease the network's complexity. It is assumed that the input image size is $200 \times 200$, and the number of neurons in the next layer is 200. When the image is processed by full connection, the required weight parameter is $200 \times 200 \times 200 = 8 \times 106$. When the local connection is applied, it is presumed that the local receptive field is $5 \times 5$, and the weight parameter is $5 \times 5 \times 200 = 5000$. So the weight parameters are reduced by 1600 times. When the input picture size is larger, the effect of lowering the number of parameters is more obvious. Local connection can also effectively avoid overfitting.

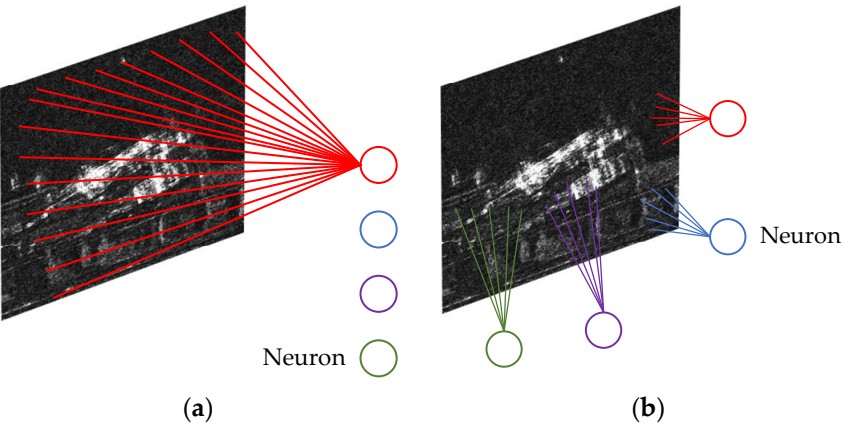

**Figure 4.** Fully connected and locally connected: (**a**) Fully connected sketch map; (**b**) Locally connected sketch map.

### 4.1.2. Weight Sharing

The values in the filter are called weights. Therefore, the so-called weight sharing refers to the convolution operation of the entire image with an identical filter. The values do not change as the position changes in the image. For CNN, the filter is generally named convolution kernel. Weight sharing is only for neurons at the same depth, and the neuron weights at different depths are not shared. Weight sharing has two functions. First, it can extract the same features at different locations in the same image. Second, it could considerably decrease the number of training parameters. For the locally connected network, the weight parameters are not shared. The comparison between the two approaches is shown in Figure 5. By using weight sharing, CNN can considerably reduce the difficulty of network training and achieve parallel training. At the same time, it can also improve the model's generalization ability.

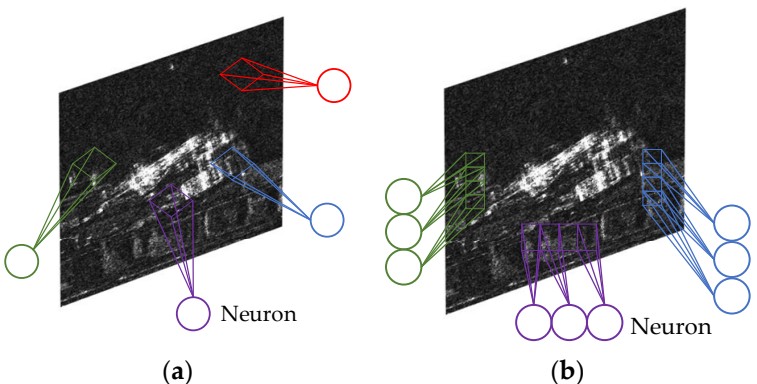

**Figure 5.** Locally connected network and weight sharing: (**a**) Locally connected network; (**b**) Weight sharing.

### 4.1.3. Sampling Layer

In CNN, the layer is mainly executed by a pooling operation, so the sampling layer is also called the pooling layer. The sampling layer takes advantage of the similar local statistical characteristics of the image. The lower-level local features are aggregated into higher-level features to fully characterize the input image. The input of the sampling layer generally comes from the output of the previous convolutional layer. The sampling layer can compress the number of data and parameters, enhance the model's robustness, and reduce overfitting.

Up to the present, convolutional neural networks have evolved into numerous different structures, but their basic structures have not undergone major changes. In the basic structure, the network comprises a convolution layer, a pooling layer, and a fully connected layer. The basic structure of CNN is shown in Figure 6. In addition, since the nonlinear properties can make the network approximate any nonlinear mapping, the nonlinear activation in the CNN cannot be neglected. Therefore, convolution, pooling, nonlinear activation, and fully connected classification are common basic operations in CNN.

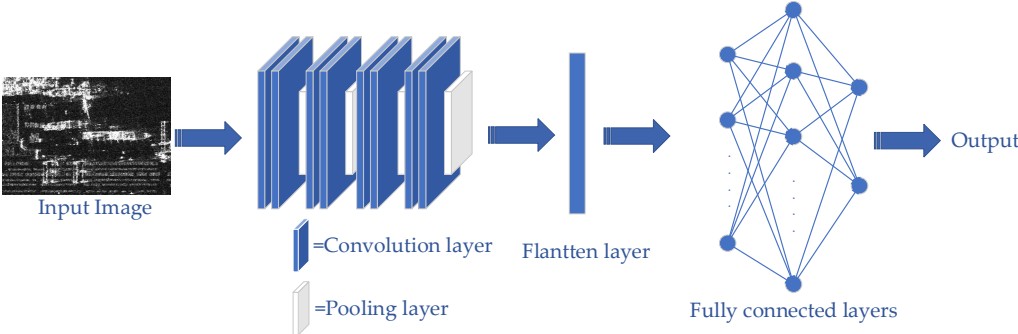

**Figure 6.** CNN basic architecture diagram.

The convolution operation is primarily to extract the image features, and feed the extracted features to the next layer for network learning. The convolution operation is completed by multiple convolution kernels. The specific procedure is to use a fixed-size convolution kernel and to traverse the entire image of the layer with a certain step size. The weight on the convolution kernel is multiplied by the corresponding position of the pixel value in the image, and then the summation operation is performed. This sum is the value after the convolution operation. After repeating the operation, a fixed-size feature map can be obtained. Figure 7 is the specific implementation process of obtaining a value on the feature map. The feature maps gained by different convolution kernels are also different. In addition, weight sharing in CNN can significantly reduce the risk of overfitting.

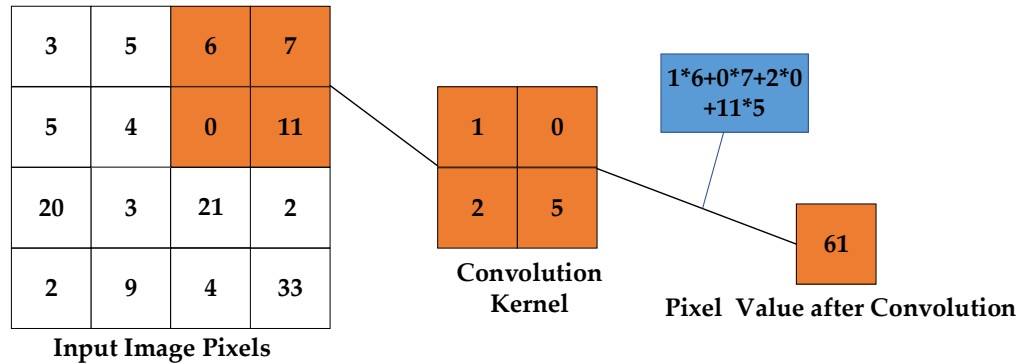

**Figure 7.** Convolutional operation diagram.

The pooling operation can also be called down-sampling. The concrete process is that on the feature map after convolution, the pooling window moves in a certain order,

and finally outputs an element of the feature map. Familiar pooling operations are global average pooling and maximum pooling, as shown in Figure 8. The pooling operation can decrease the number of parameters, speed up the program's operation, and make the model more robust as well. Common pooling window sizes are 2 × 2, 3 × 3, etc.

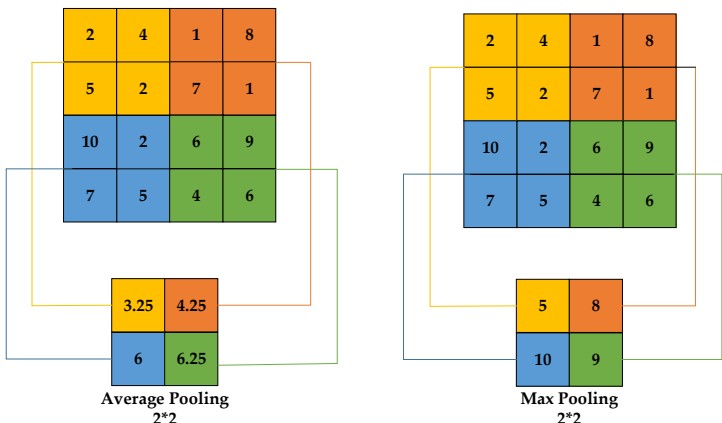

**Figure 8.** Pooling operation diagram.

The fully connected layer is composed of some interconnected neurons, which can classify and regress the input data. The classification is to classify the target or image, and the regression is mainly to regress the parameters of the bounding box.

The nonlinear operation is to introduce some nonlinear activation functions into the network. The common nonlinear functions are sigmoid, tanth, ReLU functions, etc. [24]. In these four types of nonlinear activation functions, the ReLU function can avoid gradient vanishing and make the network better training.Itcan also make CNN sparse and convergence earlier. Hence, the ReLU function is the most common function in CNN.

## 4.2. Research Progress of CNN in Optical Image Field

The CNN can be traced back to the "neocognitron" model of Japanese scientist Fukushima, K [25]. However, due to various limitations at that time, the neural network did not attract interest. In 1998, Lecun et al. proposed the LeNet-5, which was the first time that CNN was applied to digital recognition. People began to gradually apply CNN to scientific research tasks [26]. With the introduction of nonlinear activation functions and dropout, CNN has gradually attracted people's attention. In 2012, Krizhevsky, A et al. first used CNN for large-scale image classification tasks and proposed the AlexNet, which significantly boosted the accuracy of classification. Finally, they relied on AlexNet to win first place in the ILSVRC competition [27]. AlexNet's results have sparked an upsurge in CNN research and learning. In 2014, Oxford University proposed the VGG [28], which gained runner-up in the ImageNet competition. The success of the VGG network demonstrates that increasing the network's depth could vastly increase the model's accuracy and that the VGG uses small convolution kernels (3 × 3 convolutional layers and 2 × 2 subsampling layers), which can significantly enhance the network's performance. Also in 2014, Google designed the GoogLeNet [29] and won the title in the ILSVRC2014 competition. In GoogLeNet, the most significant point is the Inception module. When constructing the network structure, the author considers the network's depth as well as width. Under the premise that the number of parameters is decreased, the network's performance is upgraded, and the training efficiency is elevated. In 2016, He, K proposed the deep residual network called ResNet [30]. A new structure named "shortcut connections" was adopted in ResNet, which can solve the network's degradation problem, thus making it possible to train deep CNN. The number of ResNet's layers has reached 152. The accuracy rate of the image classification is 96.53%, and the recognition performance has surpassed the human eye.

In addition, some detection algorithms with superior performance have been proposed, such as two-stage target detection algorithms, which represent R-CNN [31], SPP-Net [32], Fast R-CNN [33], Faster R-CNN [34], etc. One-stage target detection algorithms include the YOLO [35] series, SSD [36], and so on. In current years, many researchers have gradually noticed that they cannot blindly upgrade the network's accuracy regardless of the number of parameters. We need to achieve a trade-off between the number of model parameters and the accuracy. Based on this, many lightweight CNNs have emerged, such as MobileNet [37], ShuffleNet [38], etc. These networks have been increasingly used in real-time terminal devices. Figure 9 shows some typical target object detection algorithm development, and Table 4 lists the performance parameters of some typical object detection algorithms. The details are as follows.

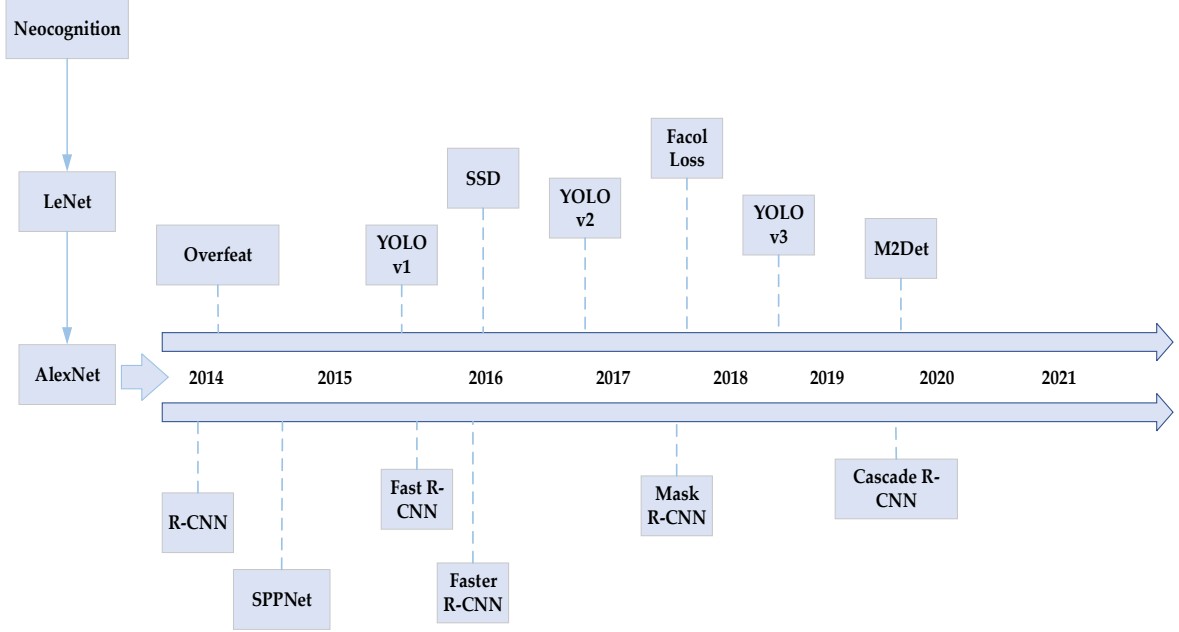

**Figure 9.** The development flow chart of typical target detection algorithms.

**Table 4.** Performance parameter table of typical target detection algorithms.

| Target Detection Algorithm | Backbone Network | Image Size | mAP (VOC) | mAP50 (COCO) | FPS (TitanX) |
|---|---|---|---|---|---|
| YOLOv1 | 24-layer convolution | 448 × 448 | 63.4 | - | 45 |
| YOLOv2 | Darknet-19 | 416 × 416 | 76.8 | 44.0 | 67 |
| YOLOv3 | Darknet-53 | 416 × 416 | - | 55.3 | 78 |
| SSD300 | VGG-16 | 300 × 300 | 74.3 | 41.2 | 46 |
| SSD512 | ResNet-101 | 512 × 512 | 76.8 | 46.5 | 19 |
| Faster R-CNN | VGG-16 | 1000 × 600 | 73.2 | 42.7 | 7 |

## 5. SAR Image Research

### 5.1. SAR Image Detection and Processing

#### 5.1.1. SAR Image Dataset

For the past few years, SAR satellites have been launched all over the world, which has significantly promoted the progress of SAR image research. Based on this, many experts and scholars have constructed some SAR image data sets. These data sets contain more and more target types and image scenes, which are beneficial to the progress of SAR image research. The following five types of datasets are more common in SAR images: including the SAR ship detection dataset (SSDD), the High-Resolution-SAR Images (HRSID), the

SAR-Ship-Dataset, OpenSARShip and MSTAR (Moving and Stationary Target Acquisition and Recognition).

SSDD [39] was constructed by Professor Li, J. The dataset contains 1160 pictures and 2456 targets, with an average of 2.12 ships per image. SSDD is the first dataset for SAR target detection. In this dataset, the images come from three different satellite sensors. The dataset has imaging pictures of four polarization modes. The resolution of the image is about 1~15m, and most of the target ships are distributed in different scenarios of nearshore or offshore. Besides, SSDD is labeled by an opensource software called LabelImg, which improves considerably the accuracy of labeling ships.

HRSID [40] contains 5604 pictures with 16,951 targets. The size of each picture is 800 pixels × 800 pixels. The dataset uses the most advanced verifier and uses the MS COCO dataset's annotation format to make the image's label. In order to guarantee high-quality imaging, the author chose the high-resolution imaging mode of the satellite when building the dataset. In the process of cutting the image, the offshore area and the ship-intensive area are separated, separately. For the single target in the far sea, a custom threshold is used, and 20% is used as the repetition rate of the cutting. The sliding window is 800 pixels × 800 pixels. Finally, when marking, the ship target in the image is marked in a polygonal manner, and the final file is saved in JSON format.

The SAR-Ship-Dataset [41] consists of two different image types: 102 GF-3 images and 108 Sentinel-1 images. The SAR-Ship-Dataset contains 43,819 ship slices, and the size of each image is 256 pixels × 256 pixels. These ships have different scales and backgrounds, which increases the target randomness. The dataset has many complex background ship targets, which provides a possibility to promote the algorithmic robustness, and can also improve the generalization algorithm performance. In the process of classifying image data, the whole dataset is randomly split into a training set, validation set and test set on the basis of the distribution ratio of 7:2:1. The image samples of the dataset are more numerous, so the target detection model can learn more abundant image features, which perfects the model's accuracy, and has a certain contribution to enhancing the model's detection performance.

OpenSARShip [42] is a kind of data set that comprises target types. The dataset was created by Shanghai Jiaotong University in 2017. In terms of satellite selection, the images are from the Sentinel-1A satellite. There are about a dozen types of ships in the dataset, with cargo and tanker being the largest number of them. The dataset contains approximately 10,000 SAR ship image slices. These image slices are from 41 Sentinel-1A SAR pictures. According to the different polarization methods, the slice data can be split into two categories: VH polarization and VV polarization. According to the different imaging modes, the data set can be divided into ground range detected (GRD) and single look complex (SLC). The image resolution in these two modes is 20 m × 20 m, 2.7 m × 22 m~3.5 m × 22 m, respectively. Unlike GRD mode, SLC mode also contains phase information.

The MSTAR dataset [43] comes from the Moving and Stationary Target Acquisition and Recognition program in the United States, which is a dataset of SAR ground stationary targets. The dataset is collected by high-resolution spotlight SAR, and the image size is 128 pixels × 128 pixels. Static SAR vehicle slices occupy the vast majority of the dataset. In the MSTAR dataset, there are not only stationary SAR vehicle images but also some environmental scene data. These scene data are obtained by SAR in strip mode, and their sizes are not exactly identical. Meanwhile, the MSTAR dataset plays a vital role in the pre-training of the model. The dominant cause is that SAR images are mostly grayscale images, which are different from RGB images. Therefore, using the MSTAR dataset for pre-training can avoid negative migration.

Tables 5 and 6 represent the information table of the SAR ship dataset and ten types of target recognition problems in MATAR images, respectively. Details about the above portion data sets are as follows.

**Table 5.** Information table of SAR ship dataset.

| Dataset | Height | Width | Sample Size | Target Quantity | Average Height/Pixel | Average Width/Pixel | Average Area/Pixel |
|---|---|---|---|---|---|---|---|
| SSDD | 190~256 | 214~668 | 1160 | 2540 | 39.05 | 36.08 | 1882.89 |
| HRSID | 800 | 800 | 5604 | 16,951 | 33.16 | 37.65 | 1809.72 |
| SAR-Ship-Dataset | 256 | 256 | 43,819 | 59,535 | 33.32 | 31.32 | 1133.88 |

**Table 6.** Ten types of target recognition problems in MATAR images.

| Category | Training Set of Pitch Angle 17° | Test Set of Pitch Angle 15° |
|---|---|---|
| BMP2(sn-9566) | 233 | 196 |
| BTR70(sn-c71) | 233 | 196 |
| T72(sn-132) | 232 | 196 |
| BTR60 | 256 | 195 |
| 2S1 | 299 | 274 |
| BRDM2 | 298 | 274 |
| D7 | 299 | 274 |
| T62 | 299 | 273 |
| ZIL131 | 299 | 274 |
| ZSU23/4 | 299 | 274 |
| Summation | 2747 | 3203 |

According to the above five types of data sets, MSTAR and OpenSARShip are data sets with target type information, and the other three are data sets without target type information. Compared with other public datasets, the MSTAR dataset has a higher resolution and an earlier time to open. Therefore, the MSTAR dataset is the most widely applied data set in SAR image target detection. The first is the sample expansion of the MSTAR dataset. Song, Q et al. [44] used generative adversarial networks and adversarial auto-encoders to enhance the MSTAR dataset. In addition, the related research about the MSTAR dataset also includes the improvement of CNN, the research of transfer learning, and so on. The improvement of CNN is the main research direction. In the ship target detection dataset, SSDD is a kind of data set that was published earlier. Since SSDD belongs to the data set without target category information, the research on this data set is mainly in two directions: ship target detection and target segmentation. Nowadays, most researchers use SSDD to evaluate the proposed model, which can be found in the simulation experiments in some references. In the follow-up study of SSDD, researchers also labeled the target position after rotation, making the labeled information more accurate. However, SSDD also has a drawback, that is, the amount of data is too small. It is prone to overfitting when training the model directly by using SSDD. Therefore, in practice, it is generally used in combination with other data sets to make the model perform better. Compared with MATSR and SSDD, SAR-Ship-Dataset and HRSID were published relatively recently, so there are few studies on this data set. For the SAR-Ship-Dataset, its production team studied this dataset. Reference [45] proposed to take advantage of the RetinaNet for SAR object detection and used a feature pyramid structure to extract multi-scale features. For HRSID, Reference [46] proposed to generate simulated SAR images through sample migration and data migration, which increases the amount of the dataset and the complexity, and it was applied in SAR target detection assignments. Finally, for OpenSARship, the research focus is object recognition based on semi-supervised learning. In [47], the author proposed a semi-supervised learning method based on a generative adversarial network, which effectively solved the over-fitting problem of complex networks caused by the small number of labeled target samples. The authors used 80%, 60%, 40%, and 20% of the labeled data in the dataset for experiments. Compared with the previous random initialization method, the results show that the accuracy is increased by 23.58%.

### 5.1.2. SAR Image Preprocessing

In SAR image target detection, data preprocessing is an inevitable operation, which can increase the algorithmic accuracy. The most basic operations of preprocessing are image denoising and data enhancement. Speckle noise is the main noise in SAR images, and the primary reason is the effect of the SAR imaging mechanism [48–50]. Therefore, it is necessary to suppress or eliminate this kind of noise for SAR image denoising. According to previous studies, the denoising algorithms can be divided into the following three aspects:

1. Denoising algorithms based on spatial filtering. They mainly include Lee filtering [51], Frost filtering [52], and Non-Local-Mean (NLM) denoising [53];
2. Denoising algorithms based on transform domain. They principally include wavelet domain SAR image denoising [54], shearlet domain SAR image denoising [55], and contourlet domain SAR image denoising [56];
3. Recently, with the rapid progress of deep learning (DL), image-denoising algorithms based on the DL have gradually been favored by researchers. They have been diffusely applied and achieved nice results [57–59].

Due to the characteristics of SAR image imaging, it is often impossible to have a well-labeled large-scale SAR image dataset. For SAR image target detection, especially for CNN-based target detection algorithms, the lack of SAR image datasets is often a vital factor restricting algorithmic development. As a consequence, data expansion for small sample data sets is extremely important. Based on the existing data sets, the data sets are significantly expanded by changing the image pixels, image transformation, or noise disturbance. Finally, the neural network is trained by the expanded dataset and the original dataset simultaneously, which could greatly improve the network's performance, and increase the detection rate as well as reduce the false alarm rate. Table 7 shows some common operations for expanding datasets [60].

**Table 7.** Some common data enhancement technologies.

| Name | Title Principal Method |
| --- | --- |
| Rotation, flip, zoom, pan | To rotate an image at a certain angle, flip, zoom in or out, or shift in a plane |
| Reflexive transformation | Axial reflection transformation, specular reflection transformation of images |
| Scale transformation | Scale the image according to the scale factor to adjust the blur degree of the image |
| Noise disturbance | Adding some noises such as exponential, salt, pepper, and Gaussian noise |

### 5.2. Research on SAR Image Target Detection Based on CNN

The main goal of SAR target detection is to find out the potential targets and mark the specific location in the image. Because the deep learning network could automatically and effectively learn the target invariant features via a large data set, the cumbersome process of manually extracting features can be eliminated. So the network model has favorable robustness and reduces the influence of human factors. Therefore, in SAR image target detection, compared with the conventional target detection algorithm, the target detection algorithm based on CNN has natural advantages.

At present, the research hotspots of CNN-based SAR image target detection primarily concentrate on the following five aspects:

1. Target detection in complex scenes, improve detection accuracy, and reduce false alarm rate or missed detection rate;
2. Aiming at the shortage of existing data, transfer learning and small sample learning methods are developed;
3. Real-time model detection and lightweight network;
4. Multi-scale small target detection;
5. Combination of traditional detection algorithms and deep convolutional neural networks.

On the basis of summarizing the abundant literature, the following will analyze and discuss the research status of CNN-based SAR image target detection from the above five aspects.

### 5.2.1. Target Detection in Complex Scenes

Due to the particularity of the SAR imaging mechanism, some background clutter will inevitably occur in the imaging process. These background clutters will have an adverse influence on the SAR target detection, which will easily decrease the algorithmic accuracy and increase the false alarm rate. In response to this issue, many experts and scholars have conducted in-depth research based on the idea of CNN and achieved remarkable results. Xiao, Q proposed a multi-resolution target detection algorithm in Reference [61], which could effectively and accurately detect targets in multi-resolution SAR images, especially in complex backgrounds. For SAR image target detection in complex scenes, Yue, B [62] designed a feature extraction network based on VGG and dilated convolution, which could significantly increase the detection speed and network's accuracy. Aiming at the situation of missed detection and false detection under complex background, Xue, Y et al. [63] improved the SSD based on the knowledge of the fusion attention mechanism. Experiments show that compared with the initial SSD, the model's average accuracy is increased by 4.2%, and its anti-interference ability is also improved. In [64], in order to solve the issue of clutter interference to the detector in complex scenes, the author proposed a SAR target detection algorithm based on a fully convolutional neural network (FCN). The core idea is to convert the target detection problem into the classification of image pixels. The test results exhibited that the algorithm could effectively decrease the false alarm target and upgrade the detection performance as well as anti-interference ability.

One of the difficulties in target detection under complex scenes is detecting targets near the coast, such as ports and docks. The SAR images in the case of open sea and offshore are shown in Figure 10. Offshore areas are close to land, which vastly increases the complexity of the background. Therefore, the requirements for the detection model are further improved in this case. In [65], Fu, X et al. proposed a near-shore SAR object detection algorithm (SC-SSD) based on scene classification. The algorithm can accomplish better detection results in the case of more land scenes, and its detection speed is also significantly enhanced. Aiming at the difficulty of correctly identifying near-shore ships and land targets in the SAR images, Liu, L [66] proposed a new sea-land segmentation method, which used a multi-scale fully convolutional network (MS-FCN) as a foundation, and applied the target detection method based on rotating bounding box (DRBox) to offshore ship detection. Because this method combines the SAR images' global information and local information, it has high detection accuracy. Experiments show that this method can successfully locate most offshore ships. In References [67,68], some solutions have also been proposed for target detection in complex backgrounds. Good results have been achieved.

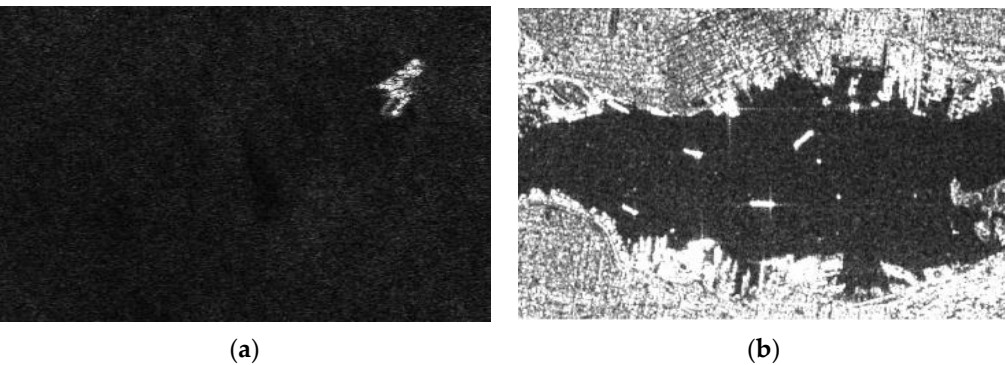

| (**a**) | (**b**) |

**Figure 10.** SAR images under two different sea conditions. (**a**) Shows the SAR image of the far-sea condition; (**b**) Shows the SAR image of the offshore sea condition.

### 5.2.2. Transfer Learning and Small Sample Learning Methods

It is well known that the target detection algorithm based on CNN has a relatively effective detection performance and powerful feature extraction ability. Nonetheless, the premise that the network model has this kind of ability is that it requires to be supported by a substantial amount of image data. Yet for SAR image data, it is usually challenging and costly to gain a substantial number of images, especially for SAR images with labels. Based on this, the introduction of transfer learning and small sample learning methods in SAR target detection is particularly significant.

Transfer learning refers to transferring a network to a learning task with a small amount of data after it has been fully trained on a large data set. Transfer learning is widely used in learning tasks with insufficient training data. Aiming at insufficient data in the SAR image dataset, transfer learning has been favored by many scholars. In Reference [69], based on the idea of fine-tuning in transfer learning, the author first pre-trained ResNet101 on the MASAR dataset and then fine-tuned the network by using the SSDD, which effectively improved the algorithmic convergence speed and robustness of the algorithm. Based on this, Li, Y et al. [70] also pre-trained ResNet on the PASCAL VOC 2007 dataset and then used the pre-trained weights to initialize and fine-tune the weights. The results showed that algorithmic detection accuracy reached 94.7%, and achieved satisfactory results. Reference [71] adopted a two-stage transfer learning method of model fine-tuning and intra-batch balanced sampling, which effectively solved the problem of unbalanced data in SAR images.

Aiming at insufficient SAR image data, in addition to network transfer learning, data expansion is also a functional method. As mentioned above, the general dataset expansion is mainly intended for making changes on the basis of existing images, such as mirroring, translation, and flipping, for example in [72,73]. This method is relatively powerful in implementation and can play the role of data expansion. However, if a mass of data sets needs to be expanded, it is not significant to improve the model's performance by using this method alone. Recently, the rapid development of a generative adversarial network (GAN) [74] provides a new idea for expanding data sets. The schematic diagram of GAN is shown in Figure 11. In the process of network training, the generator generates some images to deceive the network, and the discriminator is responsible for determining whether the data is true data, which is essentially a dynamic network game process. The use of GANs can simulate well the distribution characteristics of the original data set and can effectively enlarge the dataset. In [75], a GAN-based SAR image data enhancement method was proposed. This method uses a gradient penalty WGAN (Wasserstein GAN) to generate new samples based on existing SAR data, which can increase the number of samples in the training dataset. Compared with the traditional linear data generation method, the proposed method significantly improves the quantity and quality of training samples, and can effectively solve small sample recognition. In addition, Guo, Y [76] proposed an adaptive Faster R-CNN detection algorithm based on the knowledge of general optical images and combined it with the GAN to constrain. Simulation experiments show that this method can learn with effect and train small sample data sets, and has better performance than conventional Faster R-CNN.

The above two data augmentation strategies have different characteristics. For such measures as random cropping, the outstanding advantage of this method is that the execution process is more convenient. It also does not need to design or use other complex models. For the data augmentation strategy using GAN, this process is more complicated, and a GAN needs to be trained. Sometimes we also need to further improve GAN processing, which is more troublesome. Furthermore, the training of GAN has high hardware requirements. However, from the effect of data enhancement, the data generated by GAN is better than random cropping. For example, in [73,75], ablation experiments in [73] showed that using data augmentation techniques such as random pruning can increase the network accuracy from 68.8% to 69.6%. In [75], data generated using GAN increased the

detection accuracy from 79% to 91.6%. According to these two ablation experiments, it can be preliminarily concluded that the simulation data generated by GAN is better.

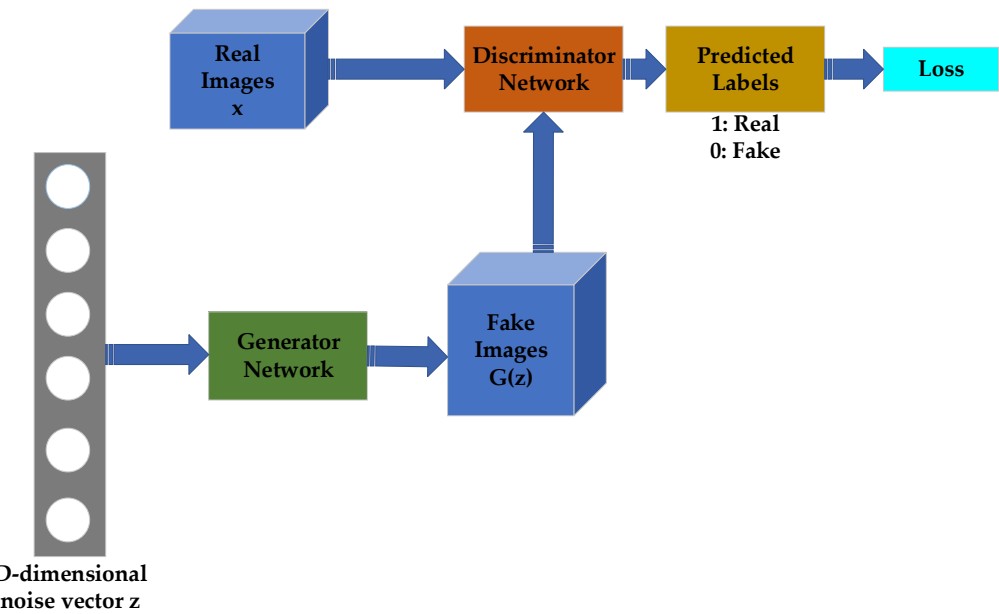

**Figure 11.** The principle diagram of GAN.

### 5.2.3. Real-Time Model Detection and Lightweight Network

In target detecting of SAR images, some scholars are committed to improving the detection model's accuracy, but this often makes the designed or ameliorative algorithms have greater redundancy. These algorithms rely heavily on computing power, and it is difficult to achieve real-time detection requirements on the terminal. It is difficult to extensively promote it in practical applications such as real-time maritime monitoring, maritime rescue, and emergency military planning. It is not desirable to sacrifice the detection speed in order to gain higher precision. Therefore, it is necessary to further explore the detection speed and achieve a favorable trade-off between detection speed and accuracy.

To enhance the detection speed of the detection algorithm, it can be expanded from the following aspects:

One is based on the original target detection algorithm to improve. The main meliorative algorithm is based on the single-stage target detection algorithm. This is because compared with the two-stage target detection algorithm, the single-stage target detection does not extract the candidate box, which can greatly boost the detection speed. In [77], Chang, Y.-L proposed a real-time target detection system. The system uses YOLOv2 as a deep learning framework. In order to decrease the computational time and increase the detection accuracy, a new structure, named YOLOv2-reduced, is developed. The detection results in SSDD and DSSDD are 90.05% and 89.13%, respectively. Compared with Faster R-CNN, the accuracy is improved and the computation time is significantly reduced. In [78], the author also proposed an improved algorithm based on YOLOv3, which used Darknet-19 as the feature extraction backbone network and reduced the network size of traditional YOLOv3 for this specific ship detection task. The results prove that the modified algorithm has a faster detection speed and the precision is basically unchanged. References [79,80] are also based on this idea to enhance the speed of the target detection algorithm and these have attained effective results. Thus, in increasing the speed of target detection, the YOLO series algorithm is widely used.

The second is the idea of using lightweight networks. In order to make a complex network lightweight, it is typically essential to start from the following three aspects. One is to compress the trained model, such as knowledge distillation and network pruning; the

second is to directly train a lightweight network, such as MobileNet and ShuffleNet; the third is improvements in hardware deployment. Based on the thinking of the knowledge distillation, a light ship detection algorithm named Tiny YOLO-Lite is proposed in [81]. The author enhanced the channel-level sparsity of the backbone network through network pruning and used knowledge distillation to make up for performance degradation caused by network pruning. In addition, an attention mechanism was added. The simulation results reveal that although the size of the network model is only 2.8M, its detection speed has exceeded 200 fps, which significantly boosts the detection speed and upgrades the model's performance. Unlike [81], Zhou, L [82] proposed a lightweight CNN called LiraNet, which combined dense connection, residual connection, and group convolution. Based on this, a Lira–you-only-look-once (Lira-YOLO) network model was proposed, which could be easily deployed on mobile devices. The experimental results show that the complexity of the Lira-YOLO network is very low, and the number of parameters is relatively small. Simultaneously, Lira-YOLO has better detection accuracy.

In addition to the above two methods, Zhang, T [83] made full use of deep separable convolution (DS-CNN). The authors integrated a multi-scale detection mechanism, cascade mechanism, and anchor frame mechanism, and used DS-CNN instead of traditional CNN. The above operations can reduce the number of network parameters immensely and increase the detection speed. Experiments on SSDD verify the correctness and feasibility of the proposed method. Meanwhile, the network also has a strong migration generalization ability.

### 5.2.4. Multi-Scale Small Target Detection

In most cases, small targets in SAR images are caused by the small size or low resolution of the target itself. Since small targets occupy fewer pixels and SAR images often contain a lot of background clutter, this undoubtedly brings many difficulties to SAR image target detection. The existence of small targets in SAR images is the main factor in missed detection.

In order to enhance the performance of multi-scale small target detection, the key is to ensure that small targets will not lose information in high-level features. In CV, the common method for small target detection is feature fusion, that is to say, the location information of the underlying features and the semantic information of the high-level features are adequately fused. Only by learning the fused information can the network have better multi-scale detection ability. The idea of feature fusion is specific to the network algorithm, which is Feature Pyramid Networks (FPN) [84]. Figure 12 shows the specific structure of the FPN. References [85–87] have adopted the feature pyramid fusion method to detect small targets and achieved satisfactory results. Based on YOLOv3, Hu, C et al. [73] introduced the design idea of a residual network as well as a feature pyramid structure and also introduced a class of balance factors, which can effectively optimize the weight of small targets in the loss function. The results demonstrate that the algorithm has better detection performance for small targets. Reference [88] introduced the attention mechanism into the target detection algorithm of multi-scale feature fusion, which significantly improved the detection performance of small targets. In order to solve multi-scale target detection in spaceborne SAR images, Liu, S et al. [89] put forward a new detector called Receptive Field Block (RPF). RPF adds dilated convolution and uses four residual structures to connect the input and output of the branch. In addition, the author also thoroughly considered the effect of the parameters on the model's performance, replacing the original $7 \times 7$ convolutions with $1 \times 7$ and $7 \times 1$ convolutions, which significantly decreases the model's complexity. Experiments on the SSDD reveal that the model's mAP reaches 95.03%. The detection speed increased to 47.16 FPS, and the model size also decreased significantly. Aiming at the poor sensitivity of the model to different ship scales in ship detection, Cui, Z et al. [90] proposed a dense attention pyramid network (DAPN) based on the FPN. The structure makes full use of the CBAM module to completely connect the bottom and top features of the feature pyramid. This method extracts rich features containing resolution and semantic

information and solves the problem of multi-scale ship detection. The simulation results show that this method has extremely high detection precision, but the model has poor adaptability to different scenarios. Further improvement and research are needed for this problem.

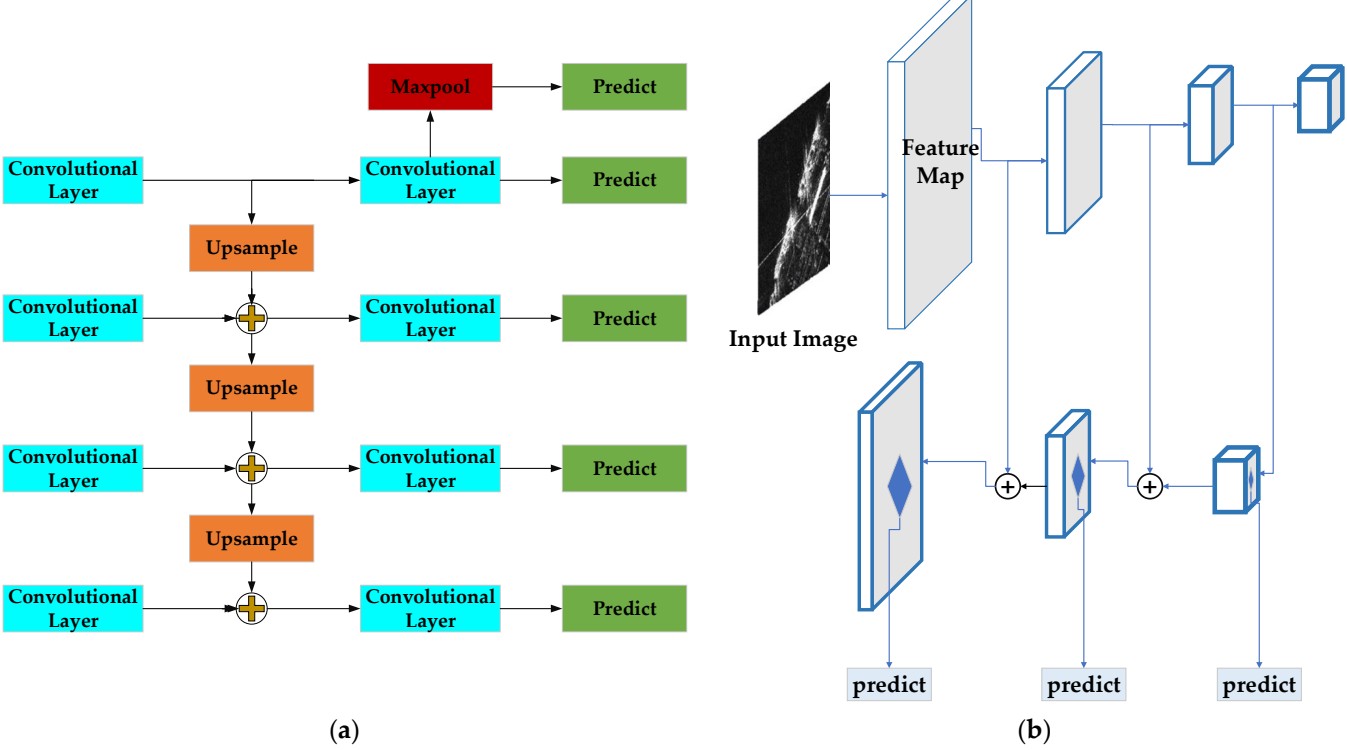

**(a)**                                                                                       **(b)**

**Figure 12.** Typical structure of characteristic feature pyramid network. (**a**) A schematic of the FPN, including convolution, upsampling, as well as maxpooling operation, and so on; (**b**) Represents the topology diagram of FPN. In this figure, white squares indicate feature maps, and thicker outlines denote semantically stronger features.

### 5.2.5. Combination of Traditional Target Detection Algorithm and CNN

In the traditional SAR target detection algorithm, CFAR is the earliest and most mature. Its computational complexity is also lower than that of CNN. Therefore, the combination of CFAR and CNN can decrease the number of parameters and increase the detection speed. Reference [91] proposed a ship-borne detection method based on traditional CFAR and lightweight DL. The experimental results show that the detection algorithm has a high detection speed and basically achieves the effect of real-time detection. Cui, Z et al. [92] proposed a constant false alarm rate (CP-CFAR) detection algorithm with convolution and pooling. The convolution layer in this algorithm uses horizontal and vertical Sobel operators to improve the contrast between the target and the background. The pooling layer reduces the processing dimension of the image. Adding a convolution layer and a pooling layer before a two-parameter CFAR can lessen computational elements without losing the main features of the original image. The simulation results show that the algorithm has fast detection speed and the running time is less than 192ms. Aiming at low detection accuracy of multi-scale ship targets, a target detection algorithm combining Faster R-CNN and CFAR was proposed in [93]. The algorithm uses Faster R-CNN to generate different sizes of regional proposal boxes. For some low-confidence proposal boxes, let CFAR detect them. Finally, the detection results with high confidence and the CFAR detection results are taken as the final output. The simulation results demonstrate that the algorithm can effectively resolve the problem of multi-scale target detection. However, for some small targets, its detection performance is poor, so the algorithm needs to be further improved for this problem.

The above experimental studies all reflect excellent CNN's performance in SAR image target detection. According to the above-related literature, the detection method and detection effect of some algorithms are obtained, as shown in Table 8.

**Table 8.** Some detection algorithms and detection effects of CNN in SAR image target detection.

| Algorithm Improvement | Specific Ideas | Pending Problems | Algorithm Effect |
|---|---|---|---|
| Algorithmic Optimization | Integrating attention mechanism | SAR target detection in complex background | Ref. [63]: high accuracy and strong robustness |
| | Converting target detection to pixel classification | The interference problem of scene clutter | Ref. [64]: lower FAR and better universality |
| | Combining scene classification | Land-sea segmentation | Ref. [65]: better detection accuracy, faster speed |
| | Lightweighting the network with knowledge distillation | Algorithm detection speed and algorithm redundancy | Ref. [81]: minor algorithmic size, faster detection speed |
| | Combining dese and residual connection, cluster convolution | Model real-time detection | Ref. [82]: low algorithmic complexity, small parameters, high accuracy |
| Expansion of the Dataset | Enhancing datasets by using GAN | Insufficient SAR image data | Ref. [75]: effectively detects small targets, and expands the dataset |
| Multi-feature Fusion | Combining adaptive anchor box algorithm and GDAL | Small target detection in spaceborne SAR images | Ref. [86]: high speed and low missed detection rate |
| | Introducing ResNet and FPN | Multi-scale small target detection | Ref. [73]: faster detection, better real-time performance |
| | Introducing attention mechanism and feature noise reduction | Multi-scale small target detection | Ref. [88]: meets the requirements of real-time detection |
| Combination of CNN and Traditional Algorithm | Combining CFAR with convolution and pooling | Long detection time of traditional algorithm | Ref. [92]: higher detection efficiency and shorter running time |

Based on the above literature analysis, it can be concluded that the dominating idea of CNN-based SAR object detection is to improve the original CV algorithm. In allusion to different problems, different algorithms need to be applied. For SAR target detection in complex scenes, the difficulty is the interference of the scene clutter to target detection. The existence of background clutter makes the false alarm rate higher, which requires that the algorithm should have better robustness. For the detection of dense small targets, feature fusion is one of the more feasible schemes. When detecting small targets in SAR images, the down-sampling of CNNs will cause the loss of some small target information, which cannot be transmitted to the deep neural network. Due to the existence of feature fusion, the underlying small target information will be better fused with the deep semantic information. As a result, the network can learn more feature information about the image. Of course, for small target detection, not only features fuse but also other methods can be used. The deepening of network layers is conducive to obtaining detection information of small targets. Therefore, the width of CNN can be increased while the network depth can be reduced, such as the Inception module. This is also a more feasible solution. To increase the model's detection speed, the key is to decrease the parameters of the model, but the prerequisite is that the model's detection accuracy cannot be reduced too much. Real-time detection of SAR image targets is also very important, which requires reducing the complexity of the model so that the model can be effectively operated at the device terminal. Unlike ordinary RGB optical image target detection, the lack of effective data sets in SAR images is one of the significant factors that hinders the application of CNNs in this field. All in all, for SAR image target detection, we need to adopt different strategies for different problems, while taking into account the impact of different factors, to enable SAR image target detection progress and development.

### 5.3. Summary of Research Status

According to the above references, as well as the research on some detection algorithms based on traditional algorithms in Section 3, after a comprehensive comparative analysis of these four types of target detection algorithms, we can present Table 9.

**Table 9.** Comparison table of SAR image target detection algorithms.

| Algorithm Classification | Advantage | Disadvantage |
| --- | --- | --- |
| Target detection algorithm based on structural feature | Fine stability, fast detection speed | Needs prior information, easy to be affected by clutter |
| Target detection algorithm based on gray feature | Nice stability, easy implement | Needs prior information, difficult to establish a unified target statistical model |
| Target detection algorithm based on image texture feature | High accuracy | Poor robustness, difficult to extract features |
| Target detection algorithm based on CNN | High accuracy and fast detection | Strong dependence on the sample, higher requirements of hardware and computing power |

Based on the above literature analysis, the four target detection algorithms have their own merits. The target detection algorithm based on structural features, compared with the other three types of target detection algorithms, has better stability, stronger robustness, and a slight advantage in detection speed. However, its apparent deficiency is the need for prior information. It can be said that the correctness of prior information determines the subsequent detection effect. Therefore, such algorithms rely heavily on prior information. Meanwhile, in some complex backgrounds, especially in near-shore target detection, this algorithmic detection accuracy is often relatively low. The target detection algorithm based on gray features has the outstanding advantage of effective stability and is not easily interfered by background clutter. Nevertheless, similar to the detection algorithm based on structural features, this kind of algorithm still needs prior information, which will limit its large-scale promotion and use of algorithms. At the same time, this kind of algorithm has difficulty in establishing a unified target statistical model and hard to achieve real-time processing in the face of a large number of image data. The detection efficiency is also low. The target detection algorithm based on texture features, compared with the first two detection algorithms, has higher detection accuracy, but this kind of algorithm is time-consuming when dealing with texture feature extraction, so the timeliness is poor. Moreover, when some dense targets are detected (such as the detection of dense ships at sea), the targets are dense and the distance between the targets is relatively close, so this will affect the calculation of the extended fractal, and then some targets will be missed. Therefore, this kind of algorithm is not suitable for detecting dense small targets. Finally, the target detection algorithm based on CNN is analyzed. The target detection algorithm based on CNN has the advantages of high accuracy and fast detection speed. However, its shortcomings are also obvious. The premise of CNN with such excellent characteristics is that it needs a substantial number of image data samples to train the network. Insufficient data is rarely seen in ordinary RGB optical image target detection. However, in the field of SAR images, based on the previous analysis, it is challenging to obtain a labeled SAR image data set with a mass of data. Therefore, in SAR target detection, insufficient data is a key factor that restricts the large-scale use of CNN in the SAR field. Meanwhile, the detection algorithm based on CNN has higher requirements for hardware GPU devices. Without a decent device, it is difficult to train a better CNN model. Therefore, the target detection algorithm using CNN should consider this limitation. The full and reasonable application of CNN in SAR image target detection still requires further research.

In general, when facing different SAR image target detection tasks, we need to choose the appropriate target detection algorithm, so that we can make full use of the advantages of various algorithms to achieve comparatively ideal results.

## 6. Future Prospects and Key Challenges

Although the SAR image target detection algorithm based on deep learning has demonstrated remarkable achievements in recent years, many problems and challenges still exist in this field, and researchers need to urgently further explore them. In the future, research work can be carried out from the following six aspects:

1.  In a complex background, the image contains substantial speckle noise, which inevitably interferes with the target detection model. Therefore, it is very important to further advance the model's robustness. In addition, the algorithmic accuracy is relatively low, which still needs to be improved. This makes it difficult to be widely used in practical fields. Therefore, in some complex backgrounds, by further improving the model's structure and training strategy, the algorithm accuracy and generalization performance can be promoted.

2.  CNN contains a considerable number of network hyperparameters. The appropriateness of hyperparameter selection substantially impacts detection accuracy. However, at present, the selection of hyperparameters for the CNN target detection network mostly relies on manual work. It is challenging to select a set of reasonable data among many hyperparameters, and artificially selected hyperparameters can easily make the detection performance of CNN worse. Therefore, the adaptive selection of hyperparameters should also be a crucial research direction in the next step.

3.  Aiming at less data and lower data complexity in the SAR image dataset, a feasible way is to use unsupervised training methods, in other words, to label a small number of samples, to train the network with unsupervised training parties. In some cases, we could directly use unsupervised training methods, and use this method to cluster. The use of unsupervised training methods can promote detection performance, so the use of unsupervised network training is also a method that cannot be ignored.

4.  Rationally design the depth of the network. Nowadays, in order to improve detection accuracy, some researchers blindly increase the depth of the network and ignore the network parameters. This situation has caused some detection models to be very bloated, and it is difficult to perform real-time detection at some terminals. Moreover, if a network model is too bloated, its training time is also very long. Therefore, it is essential to pay attention to the impact of network parameters on practical applications. In view of this situation, it is necessary to apply some ideas of the lightweight network to SAR image target detection. The lightweight network is a hot research topic in the CV field. It is an inevitable trend of SAR image target detection to consider both the detection accuracy and the amount of network model parameters.

5.  Introduce advanced CV algorithms. Nowadays, in CV, in addition to the above traditional deep learning convolution target detection algorithm, the algorithm derived from natural language processing has gradually gained more and more attention. One of the most influential is Transformer [94]. Based on Transformer, some algorithms such as Vision Transformer [95] and Swin Transformer [96] have been generated in the CV field. These two algorithms have obtained perfect results in some top competitions, indicating that the algorithm has better target detection performance in the optical field. Therefore, f making full use of some excellent algorithms can greatly improve the research status of SAR image target detection.

6.  Although at present, the detection algorithm based on CNN is developing rapidly, we cannot completely abandon those traditional target detection algorithms. The advantages of traditional target detection algorithms are still worth learning. We can fully combine the detection algorithm of deep learning with the traditional target detection algorithm to complement each other and promote the progress of SAR image target detection.

Whether SAR image target detection is based on traditional methods or based on CNN has considerably promoted the in-depth development of SAR image target detection research. For the past few years, DL has shone, which has led to the widespread use of CNN-based detection algorithms in the SAR target detection field. Currently, there are

four aspects to promote CNN's gradual engineering in the field of SAR target detection in the future:

1. Attach importance to the development of CNN to a lightweight network. We should not make the designed network very bloated in order to pursue network accuracy, that is, we should not blindly pursue the depth of the network, and should achieve a balance between the detection accuracy and speed of the network.
2. A favorable SAR image dataset is still the key factor to decide whether CNN can make full use of its advantages in the SAR field, so the construction of SAR image data set cannot be ignored.
3. Some lightweight network optimization methods in CNN are very important and worthy of further study by researchers.
4. How to make sufficient use of the self-attention mechanism in transform to maximize its own advantages and enable it to better extract the SAR image features is also a problem worth further investigation.

## 7. Conclusions

The rapid development of deep learning brings a new opportunity for SAR image target detection. As a typical kind of algorithm in deep learning target detection, the convolutional neural network has been favored by more and more researchers because of its advantages of fully mining image information, adaptively extracting target features, strong robustness, and no need for complex artificial construction features. We studied the SAR image target detection algorithm based on traditional algorithms by introducing some references and analyzed the disadvantages of traditional algorithms in solving SAR image target detection tasks. Then the necessity of using CNN to detect targets in SAR images was introduced. At the same time, we summarized a lot of the literature about the SAR image target detection algorithm based on CNN. We also discussed the contribution of the innovations in these papers to improve the detection performance of CNN and summarized the difficulties and challenges of CNN-based SAR image target detection for the future. These are the main scientific problems currently faced by using CNN to detect SAR images, including but not limited to the trade-off between CNN accuracy and detection speed, more model parameters, bloated and complex networks, and insufficiently lightweight. These problems need further research and discussion by researchers. In a word, introducing deep learning into the field of SAR image target detection is an extremely significant change of thinking, and it is also a typical example of a practical AI engineering application. It is believed that in the near future, in SAR image target detection, target detection based on CNN will attain greater success and foster continuous development.

**Author Contributions:** Y.Z. put forward the general framework of the article and provided the writing ideas, conceived and supervised the research and experiments, and contributed as the lead author of the article. For the improvement and revision of the article, Y.Z. also made many constructive comments. Y.H. consulted the references and completed the writing and revision of this article. All authors have read and agreed to the published version of the manuscript.

**Funding:** This work was supported by the National Natural Science Foundation of China (no. 61673259); supported by Shanghai "Science and Technology Innovation Action Plan" Hong Kong, Macao, and Taiwan Science and Technology Cooperation Project (no. 21510760600); and was also supported by Capacity Building Project of Local Colleges and Universities of Shanghai (no. 21010501900).

**Acknowledgments:** The authors would like to thank the College of Information Engineering and the Institute of Logistics Science and Engineering of Shanghai Maritime University for their support. The author would also like to thank the anonymous reviewers for their helpful suggestions and comments to improve the article.

**Conflicts of Interest:** The authors declare no conflict of interest.

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
