# Peer review of "A Survey of SAR Image Target Detection Based on Convolutional Neural Networks"

_remotesensing, doi:10.3390/rs14246240_

Round 1

Reviewer 1 Report

I suggest to your author to reduce the plagiarism up to 10%.

There are some questions and suggestions are proposed:

1.     The abstract needs to clarify the significance of the proposed research work.

2.     The introduction and related research of this paper is just a list of documents and their contents, lacking analysis and summary. And the literature review is not comprehensive and lacks representativeness.

3.     Lines 52-63 it will be very nice to have a reference to support the text

4.     By the way, the writing of this paper is not scientific. I suggest to your author add a separate section about related work, then methodology, and the results and discussion, I suggest restructuring the paper it will improve the quality of the paper. Please move the figure from the introduction section to some other suitable place.

5.     Please add a more detailed description of the CNN Approach, currently there are many CNN-based methods used for SAR target detection.

6.     In the methodology section, the author should describe how the author chose the paper to include and exclude. As a review paper, the article can refer to some of the latest research methods, such as Ship detection based on deep learning using SAR imagery: a systematic literature review

7. the English of the paper should be improved

Author Response

Authors' Reply Letter for AE and Reviewers

Dear Prof. Jasmine Jia and reviewers,

Thank you for your time and effort put in reviewing our manuscript entitled “A Survey of SAR Image Target Detection Based on Convolutional Neural Networks” (Manuscript ID: remotesensing-2031165). We also thank you for your detailed comments that helped us improve the quality of this paper. In the following, we address your concerns in the order that they were mentioned.

Sincerely,

Manuscript ID: remotesensing-2031165.

Reviewer #1:

Q(1): The abstract needs to clarify the significance of the proposed research work.

[Response]: Thank you and agree with your advice. Firstly, the initial abstract really has a vague expression about the significance of the proposed research work, which will make readers not understand this paper research focus. Therefore, according to your suggestion, we modified the abstract. The revised abstract elaborates on the significance the proposed research work, so that readers can have a clearer understanding of this article. The revised abstract is as follows:

‘Synthetic Aperture Radar (SAR), because of its all-day imaging, multi-band, high resolution and strong anti-weather interference capacity, possesses a very broad application prospect in some fields, such as environmental monitoring and remote sensing reconnaissance. With the development of SAR imaging technology, target detection in SAR images has become a hot research topic. Due to the poor robustness, low detection accuracy and difficulty in adjusting to complicated and changeable scenes of conventional target detection algorithms, we research the current development status of SAR image target detection based on in-depth analysis of convolutional neural network (CNN), and then envisage the future of SAR image target detection. Meanwhile, we also summarize the problems and challenges to be faced in this field, and ultimately draw some relevant conclusions. By summarizing and analyzing prior research work, this paper is helpful for subsequent researchers to rapidly recognize the current development status and identify the connections between various detection algorithms. Beyond that, this paper summarizes the problems and challenges confronted by researchers in the future, and also points out the specific content of the next research, which has certain guiding significance for promoting the progress of SAR image target detection’

Q(2): The introduction and related research of this paper is just a list of documents and their contents, lacking analysis and summary. And the literature review is not comprehensive and lacks representativeness.

[Response]: Thanks for your opinion. For the introduction of this article, we have rewritten it. The original introduction is not comprehensive enough and lacks our analysis. We modified the introduction according to your suggestion. We have deleted some redundant parts, such as paragraphs 3 and 4 in the introduction. Then we analyze deep learning in detail, summarize its characteristics, introduce its role, and gradually introduce CNN. In addition, we add the contribution in the introduction, which is mainly analyzed from four aspects. In view of the incomplete literature review, we have selected some representative references, such as [3] [4] [5] [90] [91] and so on. We also deleted some unrepresentative references. Through this revision, we believe that the article has made great progress

“[3] Lencun, Y.; Bengio, Y.; Hinton, G. Deep Learning. Nature 2015, 521, 436-444.

[4] Zhao, Z.; Zheng, P.; Xu, S.; Wu, X. Object Detection with Deep Learning: A Review. IEEE Trans. Neural Netw. Learning Syst. 2019, 30, 3212-3232.

[5] Guo, Y.; Liu, Y.; Georgiou, T.; Lew, M. S. A Review of Semantic Segmentation Using Deep Neural Networks. Int J Multimed Info Retr. 2018, 7, 87-93.

[90] Liu, S.; Kong, W.; Chen, X.; Xu, M.; Yasir, M.; Zhao, L.; Li, J. Multi-Scale Ship Detection Algorithm Based on a Lightweight Neural Network for Spaceborne SAR Images. Remote Sens. 2022, 14, 1149.

[91] Cui, Z.; Li, Q.; Cao, Z.; Liu, N. Dense Attention Pyramid Networks for Multi-Scale Ship Detection in SAR Images. IEEE Trans. Geosci. Remote Sens. 2019, 57, 8983-8997.”

Our contributions to this article are as follows:

1) For the traditional SAR image target detection algorithm, we divided the traditional detection algorithm into three categories, and studied the detection algorithm of each category with the relevant references, analyzed the basic idea, advantages and disadvantages of different algorithms under the same category. Based on this, we summarized the characteristics of these three algorithms, and then lead to the necessity of using CNN for SAR image target detection.

2) We analyzed the basic theory and network structure of CNN, and studied the SAR target detection data sets which are frequently used at present.

3) Based on a large number of references, we studied the CNN-based SAR image target detection. According to the main problems faced by CNN in SAR image target detection, we divided the literature review analysis into five categories. And we summarized the innovative ideas of various improved algorithms. At the same time, we compared CNN with traditional SAR target detection algorithms, and obtained the characteristics of various algorithms.

4) The difficulties and challenges in the field of SAR image target detection were de-rived from the analysis of references, which pointed out the direction for future research.

Q(3): Lines 52-63 it will be very nice to have a reference to support the text.

[Response]: Thanks for the reminder. Indeed, this part lacks the support of references. According to your suggestion, we have modified this section and added some references to support it. This really has a good effect.

Q(4): By the way, the writing of this paper is not scientific. I suggest to your author add a separate section about related work, then methodology, and the results and discussion, I suggest restructuring the paper it will improve the quality of the paper. Please move the figure from the introduction section to some other suitable place.

[Response]: Thank you for your suggestions for the article. We modified the article according to your opinion. The final revised framework is as follows: the first is the introduction, the second is the research methodology, the third is the study of SAR target detection, the fourth is the analysis of the focus of future research, and the last is a summary of the full text. According to your suggestion, we do not insert pictures in the introduction section. Please see them in the revised version of the manuscript.

Q(5): Please add a more detailed description of the CNN Approach, currently there are many CNN-based methods used for SAR target detection.

[Response]: Thanks for your opinion. In the initial paper, our introduction to CNN is indeed relatively simple. According to your requirements, we describe the more details of CNN in the Convolutional Neural Network section. There is no analysis of CNN’s three ideas in the original paper. For CNN, however, they are fundamental factors that distinguish them from other neural networks. Hence, we analyze them in detail. The process we analyze is the definition, structure, and advantages of each idea. We have drawn vivid pictures of local connections and weight sharing, as shown in Figures 4 and Figure 5, which can help readers better understand what we mean. Here are the images we’ve drawn and the details we’ve added.

‘1) Locally connected layer

Unlike fully connected neural networks, CNN utilizes a local connection. If each pixel in the image is regarded as a neuron, then each output neuron of the fully connected network links all the neurons in the image while CNN only links a small number of adjacent neurons in space. Figure 4 shows the specific architecture. There are two reasons for CNN to adopt local connection. Firstly, for images, local pixels are closely associated, and the correlation between pixels with farther distance is weaker. Therefore, each neuron doesn’t require to perceive the global image, but solely needs to perceive the local area. Then, the local information of low-level perception is synthesized in the high-level to obtain global information. The second reason is to reduce the number of network parameters, and decrease the network’s complexity. It’s assumed that the input image size is 200*200, and the number of neurons in the next layer is 200. When the image is processed by full connection, the required weight parameter is 200*200*200 =8*106. When the local connection is applied, it is presumed that the local receptive field is 5*5, and the weight parameter is 5*5*200=5000. So the weight parameters are reduced by 1600 times. When the input image size is larger, the effect of lowering the number of parameters is more obvious. Local connection can also effectively avoid the overfitting.

(a)

(b)

Figure 4. Fully connected and locally connected: (a) Fully connected sketch map; (b) Locally connected sketch map.

2) Weight sharing

The values in the filter are called weights. Therefore, the so-called weight sharing refers to the convolution operation of the entire image with the identical filter. The values don’t change as the position changes in the image. For CNN, the filter is generally named convolution kernel. Weight sharing is only for neurons at the same depth, and the neuron weights at different depth are not shared. Weight sharing has two functions. First, it can extract the same features at different location in the same image. Second, it can considerably reduce the number of training parameter. For the locally connected network, the weight parameters are not shared. The comparison between the two approached is shown in the Figure 5. By using weight sharing, CNN can considerably reduce the difficulty of network training and achieve parallel training. At the same time, it can also improve the model’s generalization ability.

 (a)

(b)

Figure 5. Locally connected network and weight sharing: (a) Locally connected network; (b) Weight sharing.

3) Sampling layer

In CNN, the layer is mainly executed by pooling operation, so the sampling layer is also called pooling layer. The sampling layer takes advantage of the similar local statistical characteristics of the image. The lower-level local features are aggregated into higher-level features to fully characterize the input image. The input of the sampling layer generally comes from the output of the previous convolutional layer. The sampling layer can compress the number of data and parameters, enhance the model’s robustness, and reduce overfitting.

In addition, we also add the detailed structure diagram of CNN, as shown in Figure 6, which enables readers to have a macro understanding of CNN.

Figure 6. CNN basic architecture diagram.

Q(6): In the methodology section, the author should describe how the author chose the paper to include and exclude. As a review paper, the article can refer to some of the latest research methods, such as Ship detection based on deep learning using SAR imagery: a systematic literature review.

[Response]: Thank you for your comments on the methodology of this article. As you said, we didn’t touch upon the methodology part at first. This is our deficiency. According to your suggestion and the recommended literature, we put forward the methodology section in combination with the content of this paper. We put this section in Chapter 2. In Chapter 2, we elaborate the principle of selecting references, the selection of journal database and the article research thought. The details of Chapter 2 are as follows:

” This article is a review article, so the selection of references is extremely important. From the point of the reference publication time, most of the articles are selected within the past five years, and a few articles are predominantly published between 2010 and 2015. This paper does not select literature with a longstanding gap, mainly to analyze the current research trends and summarize the latest scientific research results in recent years. In terms of selecting article journal databases, we primarily search for relevant literature from MDPI, IEEE Xplore, Science direct and other databases. For the research content, we review the SAR image target detection from two major aspects. One is the traditional algorithm, and the other is the CNN-based algorithm. Then these two large aspects are subdivided into different small aspects. For the traditional algorithms, we divide the traditional algorithms into three categories. For CNN-based algorithms, we analyze from the perspective of CNN solving different SAR detection tasks. The purpose of doing this is to make our objectives clearer, the articles more representative, and the summary more comprehensive.”

Q(7): the English of the paper should be improved

[Response]: Thanks for your reminder. We are sorry for the problem in our old manuscript. We have revised the whole manuscript and carefully proof-read the manuscript to minimize, grammatical, and bibliographical errors. In addition, we have invited a native English speaker to check the language. We believe that the language is now acceptable for the review process. Please see the revised version of the manuscript.

Please see all the modifications with the highlighted RED font in the revised version.

Thanks,

Ying Zhang

Shanghai Maritime University

2022.11.20

Reviewer 2 Report

Dear authors,

Thanks for your submission. This manuscript is a good review of SAR image target detection. The comments are as follows:

1, The shortcoming of SAR image are poor robustness, low detection accuracy and complex background.  The basement of CNN model are training sample and detecting target How to solve the problem of target recognition accuracy caused by image resolution?

2, In this manuscript, there are some typical method or structure.  Which is the best one for SAR image target detection? How about the accuracy and the efficiency?

3, In section 5, there are seven aspects for the future research.  But these are not a scientific problem with SAR image targets, and some have finished (i.e. public datasets, tools.) This is not a good summarize.

4,The conclusion is too short. What's the scientific question in SAR image target detection?  What‘s the advantages of CNN model?

5, Please double-check all reference format.

Author Response

Authors' Reply Letter for AE and Reviewers

Dear Prof. Jasmine Jia and reviewers,

Thank you for your time and effort put in reviewing our manuscript entitled “A Survey of SAR Image Target Detection Based on Convolutional Neural Networks” (Manuscript ID: remotesensing-2031165). We also thank you for your detailed comments that helped us improve the quality of this paper. In the following, we address your concerns in the order that they were mentioned.

Sincerely,

Manuscript ID: remotesensing-2031165.

Reviewer #2:

Dear authors,

Thanks for your submission. This manuscript is a good review of SAR image target detection. The comments are as follows:

Q(1): The shortcoming of SAR image are poor robustness, low detection accuracy and complex background.  The basement of CNN model are training sample and detecting target How to solve the problem of target recognition accuracy caused by image resolution?

[Response]: Sincerely thank you for your affirmation of the paper.

As mentioned in the paper, the rapid development of SAR imaging technology makes the SAR image resolution continue to improve. Therefore, we will inevitably encounter the problem of target recognition accuracy due to the improvement of SAR image resolution. As we know, the improvement of SAR image resolution will increase the type and number of targets in the image. Based on this, we believe that solving this problem is mainly analyzed from three aspects: how to deal with high-resolution input, how to improve dense small target detection, and how to solve the problem of class imbalance. Regarding how to improve the input of high resolution, the common way is to cut a high-resolution image into multiple sub-images. In order to avoid cutting to the target, we can use overlapping cutting method. For how to improve dense small target detection, as mentioned in the paper, we can use FPN structure for feature fusion. For some anchor-based target detection algorithms, we can design anchors that match the small target size. Regarding how to solve the problem of category imbalance, we can use the upsampling and downsampling of data to balance the amount of data between different categories or design new loss functions. Of course, what we summarize may not be very comprehensive, there must be other ways, and we need further research on this problem.

Q(2): In this manuscript, there are some typical method or structure.  Which is the best one for SAR image target detection? How about the accuracy and the efficiency?

[Response]: As we can see, for SAR image target detection, there are a lot of target detection algorithms, such as typical traditional algorithm (i.e. CFAR), deep learning algorithm (i.e. CNN). These algorithms have different characteristics, which limit their application scenarios. Nowadays, in SAR target detection, more and more scholars have shifted their attention from traditional algorithms to deep learning algorithms. And now many experiments show that algorithms based on deep learning can achieve better detection results. Therefore, using deep learning algorithm to deal with SAR target detection task is the general trend. According to the text, the best choice for SAR image target detection is the deep learning algorithm, which can be said that CNN is an effective method. We can learn that most scholars have done research based on YOLO, which also shows the popularity of YOLO. Moreover, a substantial number of experiments also show that this algorithm is indeed suitable for SAR target detection tasks. Therefore, we believe that YOLO series is the best choice for SAR image target detection. Its accuracy is not much different from the two-stage target detection algorithm (i.e. R-CNN), but its efficiency is much higher than the two-stage target detection algorithm, which can be summarized from the references [77], [78], [79], [82] and so on. Of course, with the introduction of self-attention in NLP, we believe that there will be greater progress for SAR target detection.

Q(3): In section 5, there are seven aspects for the future research.  But these are not a scientific problem with SAR image targets, and some have finished (i.e. public datasets, tools.) This is not a good summarize.

[Response]: Thank you for your suggestions for future research. Indeed, as you said, the SAR data set has been well expanded in SAR image target detection. Scholars have also carried out research on labeling tools and developed many useful tools. However, this is not from the algorithm perspective to analyze SAR image target detection. According to your suggestion, we modified it. First, we have deleted the second and third of the original future research. Because for the scientific problems of SAR image targets, the second and third are not very good analysis of future trends. Secondly, from the perspective of algorithm, we summarized current difficulties in selecting network hyperparameters, and proposed that we need to study the automatic selection of hyperparameters in the future. Today, neural architecture search (NAS) is a typical algorithm for solving such a problem. But we still need further research. Finally, we summarized the second future research as follows:

“2) CNN contains a considerable number of network hyperparameters. The appropriateness of hyperparameter selection substantially impacts the detection accuracy. However, at present, the selection of hyperparameters for CNN target detection network mostly relies on manual work. It’s challenging to select a set of reasonable data among many hyperparameters, and the artificially selected hyperparameters can easily make the detection performance of CNN worse. Therefore, the adaptive selection of hyperparameters should also be one of the crucial research directions in the next step.”

Q(4): The conclusion is too short. What's the scientific question in SAR image target detection?  What‘s the advantages of CNN model?

[Response]: Thank you for your suggestions.

Indeed, as you said, the conclusion of this paper is too concise, which is not a good summary. According to your suggestion, we re-summarized the conclusion. The revised conclusion first describes the advantages of CNN, such as fully mining image information, adaptively extracting target features, strong robust-ness and no need for complex artificial construction features. Secondly, we briefly analyzed the paper’s framework. Then according to the full text, we summarized some scientific problems in SAR image target detection, such as the trade-off between CNN accuracy and detection speed, more model parameters, bloated and complex networks, and insufficient lightweight. Finally, we have made a prospect to the SAR research. Compared with the old conclusion, the revised conclusion is more comprehensive. We believe this basically satisfies the summary of the main paper’s contents. The modified conclusions are as follows:

“The rapid development of deep learning brings a new opportunity for SAR image target detection. As a typical kind of algorithm in deep learning target detection, convolutional neural network has been favored by more and more researchers because of its advantages of fully mining image information, adaptively extracting target features, strong robustness and no need for complex artificial construction features. We studied the SAR image target detection algorithm based on traditional algorithms by introducing some references, and analyzed the disadvantages of traditional algorithms in solving SAR image target detection tasks. Then the necessity of using CNN to detect targets in SAR images was introduced. At the same time, we summarized a lot of literature about the SAR image target detection algorithm based on CNN. We also discussed the contribution of the innovations in these papers to improve the detection performance of CNN, and summarized the difficulties and challenges of CNN-based SAR image target detection in the future. These are the main scientific problems currently faced by using CNN to detect SAR images, including but not limited to the trade-off between CNN accuracy and detection speed, more model parameters, bloated and complex networks, and insufficient lightweight. These problems need further research and discussion by researchers. In a word, introducing deep learning into the field of SAR image target detection is an extremely significant change of thinking, and it’s also a typical example of the practical AI engineering application. It is believed that in the near future, in the SAR image target detection, target detection based on CNN will attain greater success and foster the continuous development.”

Q(5): Please double-check all reference format.

[Response]: Thanks for your reminder.

We are sorry that due to our negligence, there are some errors in the format of the initial references. According to your requirements, we have carefully modified the format of full-text references, and strictly followed the standard form. We believe that the modified reference format meets the standard requirements. Please see them in the revised version of the manuscript.

Please see all the modifications with the highlighted RED font in the revised version.

Thanks,

Ying Zhang

Shanghai Maritime University

2022.11.20

Reviewer 3 Report

The overall objective of this paper was difficult for me to follow. Because several SAR datasets were described, I would have expected there to be some analysis on these data sets and that the challenges laid out would be tied to actual performance on these datasets. Speaking of the challenges entirely in the abstract without considering the actual application is a less useful contribution to the field in my mind than showing results on particular proposed techniques. For example, how would the synthetic targets generated with a GAN compare to synthetic targets generated by the rotation/flipping methods using the cited datasets?  Is it useful to talk about the challenges of small targets without considering if the application is interested in fishing vessels or cargo ships? Also, is there information in the described data sets about ship class? Are you interested in tackling ship classification problems? Finally, is the lack of training data limited to public-domain datasets? Are there entities that have access to more comprehensive datasets?

The paper is a survey of current novel techniques for image processing which may indeed have application to SAR ship detection or other use of SAR but the techniques and domain do not seem to be strongly connected in the paper. The abstract and overall explanation of the layout of the paper should more clearly explain what the paper is contributing to the field. Some actual analysis with the described datasets could be useful, but if not then the purpose could be made more clear.  

Author Response

Authors' Reply Letter for AE and Reviewers

Dear Prof. Jasmine Jia and reviewers,

Thank you for your time and effort put in reviewing our manuscript entitled “A Survey of SAR Image Target Detection Based on Convolutional Neural Networks” (Manuscript ID: remotesensing-2031165). We also thank you for your detailed comments that helped us improve the quality of this paper. In the following, we address your concerns in the order that they were mentioned.

Sincerely,

Manuscript ID: remotesensing-2031165.

Reviewer #3:

Q(1): How would the synthetic targets generated with a GAN compare to synthetic targets generated by the rotation/flipping methods using the cited datasets?

[Response]: Thank you very much for your comments on this article. We are very sorry that this issue was not analyzed. This is our mistake. According to your suggestion, we modified this part carefully. Through comparative analysis, we found that for synthetic targets generated with a GAN, it’s more conducive to improve network performance, but its process is more complex. For synthetic targets generated by the rotation/flipping methods, it is obvious that this method is simple to use, but it does little to improve network performance compared to the previous method. We explain this part in detail, as follows:

“The above two data augmentation strategies have different characteristics. For such measures as random cropping, the outstanding advantage of this method is that the execution process is more convenient, and it does not need to design or use other complex models. For the data augmentation strategy by using GAN, the process of this method is more complicated, and a GAN needs to be trained. Sometimes we also need to further improve the GAN processing, which is more troublesome. Furthermore, the training of GAN has high hardware requirements. However, from the effect of data enhancement, the data generated by GAN is better than random cropping. For example, in [73] and [75], ablation experiments in [73] showed that using data augmentation techniques such as random pruning can increase the network accuracy from 68.8% to 69.6%. In [75], data generated using GAN increased the detection accuracy from 79% to 91.6%. According to these two ablation experiments, it can be preliminary concluded that the simulation data generated by GAN is better.”

Q(2): Is it useful to talk about the challenges of small targets without considering if the application is interested in fishing vessels or cargo ships?

[Response]: Thank you for your suggestion. This suggestion provides a new idea for our research. As you said, we only analyze the challenges of small targets without considering application. In the initial paper, our summary of small targets detection is not comprehensive enough. Based on your suggestion, we added additional analysis for small target detection. This time we did not only discuss the challenge of small goals, and the reference literature is also closely related to the actual. The small target analysis we added is as follows (in the 1st paragraph of Page 19 in subsection 5.2.4):

“In order to solve multi-scale target detection in spaceborne SAR images, Liu, S et al.[90] proposed a new detector called Receptive Field Block (RPF). RPF adds dilated convolution and uses four residual structures to connect the input and output of the branch. In addition, the author also thoroughly considered the effect of the parameters on the model performance, replacing the original 7*7 convolution with 1*7 and 7*1 convolutions, which significantly decreases the model’s complexity. Experiments on the SSDD reveal that the model’s mAP reaches 95.03%. The detection speed increased to 47.16 FPS, and the model size also decreased significantly. Aiming at poor sensitivity of the model to different ship scales in ship detection, Cui, Z et al.[91] proposed a dense attention pyramid network (DAPN) based on the FPN. The structure makes full use of CBAM module to completely connect the bottom and top features of the feature pyramid. This method extracts rich features containing resolution and semantic information, and solves the problem of multi-scale ship detection. The simulation results show that the method has extremely high detection accuracy, but the model has poor adaptability to different scenarios. Further improvement and research are needed for this problem.”

Q(3): Is there information in the described data sets about ship class?

[Response]: Thanks for your reminder.

In the initial paper, only the MSTAR dataset contains the target category. The remaining data sets don’t contain the target category. This is the deficiency of our summary. Therefore, according to your suggestion, we analyze the dataset OpenSARShip which contains the target category to make up for the incomplete search of SAR data set. The summary of the OpenSAR dataset is as follows (in the last paragraph of Page 12 in subsection 5.1.1):

“OpenSARShip [41] is a kind of data set which comprises target types. The dataset was created by Shanghai Jiaotong University in 2017. In terms of satellite selection, the images are from the Sentinel-1A satellite. There are about a dozen types of ships in the dataset, where cargo and tanker have the largest number of them. The dataset contains approximately 10,000 SAR ship image slices. These image slices are from 41 Sentinel-1A SAR pictures. According to the different polarization methods, the slice data can be divided into two categories: VH polarization and VV polarization. According to the different imaging modes, the data set can be divided into ground range detected (GRD) and single look complex (SLC). The image resolution in these two modes is 20m*20m, 2.7m*22m~3.5m*22m respectively. Unlike GRD mode, SLC mode also contains phase information.”

Q(4): Are you interested in tackling ship classification problems?

[Response]: Yes, we are. First of all, like ship target recognition, ship classification is also a very important image processing task. It has a very wide range of applications in many fields. Due to the limitation of this study, we didn’t carry out too much analysis. We are sorry for this. But through your opinions, it also provides a lot of guidance for our next research. In the future we will focus on the classification of ships, making our research more comprehensive.

Q(5): Is the lack of training data limited to public-domain datasets?

[Response]: Yes, it is. As we know, data sets are very important for CNN. In the current research on SAR image target detection, the data sets we use are mainly public data sets. Our research about data sets in some non-public fields is not deep enough. For public-domain datasets, the lack of data is still an urgent problem to be solved. In today’s SAR data sets, the number of images is small and the complexity of the data set is not enough, which is unfavorable for training CNN. This can easily make CNN overfitting. Nowadays, the development of SAR imaging technology makes the amount of SAR image data increasing. But it’s difficult for ordinary people to label a large number of SAR images, which requires researchers with certain professional knowledge to complete the labeling task. This is also an important reason for the lack of SAR image data.

Q(6): Are there entities that have access to more comprehensive datasets?

[Response]: Yes, there are. Nowadays, more and more scholars have carried out research on SAR data sets and produced very rich data sets. Part of this dataset is public and part is confidential. For those publicly available data sets, we can access them from databases such as IEEE, Web of Science, or MDPI. We can also search directly on Google to find the download address of each data set. Of course, we can also label a data set to make the data set more abundant.

Q(7): The paper is a survey of current novel techniques for image processing which may indeed have application to SAR ship detection or other use of SAR but the techniques and domain do not seem to be strongly connected in the paper. The abstract and overall explanation of the layout of the paper should more clearly explain what the paper is contributing to the field. Some actual analysis with the described datasets could be useful, but if not then the purpose could be made more clear. 

[Response]: As you said, these technologies and domain do not seem to be strongly connected in this paper. After our analysis and summary, we believe that in the initial paper, the techniques and algorithms proposed in some references do not have a strong connection with the purpose of this paper. Therefore, we deleted some references and referred to those that are more consistent with the paper’s content. For example, we deleted reference [3] from the initial paper and added [43] [44] [45]. These literatures have analyzed SAR data sets in detail. Based on these literatures, we focus on their applications in SAR target detection, such as the expansion of the data set and the research of transfer learning. At the same time, in the abstract and overall explanation of the layout of the paper, we did not explain the contribution in the initial paper. This is indeed our mistake. Therefore, according to your suggestions, in the abstract and introduction, we illustrate the contribution of this paper in the field of SAR target detection. The contents are as follows.

In the abstract, we have revised some of the content and briefly described the contribution of this paper. The contents are as follows:

“Abstract: Synthetic Aperture Radar (SAR), because of its all-day imaging, multi-band, high resolution and strong anti-weather interference capacity, possesses a very broad application prospect in some fields, such as environmental monitoring and remote sensing reconnaissance. With the development of SAR imaging technology, target detection in SAR images has become a hot research topic. Due to the poor robustness, low detection accuracy and difficulty in adjusting to complicated and changeable scenes of conventional target detection algorithms, we research the current development status of SAR image target detection based on in-depth analysis of convolutional neural network (CNN), and then envisage the future of SAR image target detection. Meanwhile, we also summarize the problems and challenges to be faced in this field, and ultimately draw some relevant conclusions. By summarizing and analyzing prior research work, this paper is helpful for subsequent researchers to rapidly recognize the current development status and identify the connections between various detection algorithms. Beyond that, this paper summarizes the problems and challenges confronted by researchers in the future, and also points out the specific content of the next research, which has certain guiding significance for promoting the progress of SAR image target detection.”

In the introduction, we make a concrete summary of this paper’s contribution. Contributions mainly include the following four aspects. The contents are as follows:

The main contributions of this paper are summarized as follows:

1) For the traditional SAR image target detection algorithm, we divided the traditional detection algorithm into three categories, and studied the detection algorithm of each category with the relevant references, analyzed the basic idea, advantages and disadvantages of different algorithms under the same category. Based on this, we summarized the characteristics of these three algorithms, and then lead to the necessity of using CNN for SAR image target detection.

2) We analyzed the basic theory and network structure of CNN, and studied the SAR target detection data sets which are frequently used at present.

3) Based on a large number of references, we studied the CNN-based SAR image target detection. According to the main problems faced by CNN in SAR image target detection, we divided the literature review analysis into five categories. And we summarized the innovative ideas of various improved algorithms. At the same time, we compared CNN with traditional SAR target detection algorithms, and obtained the characteristics of various algorithms.

4) The difficulties and challenges in the field of SAR image target detection were derived from the analysis of references, which pointed out the direction for future research.

Finally, as you said, in the initial we only introduced the SAR data sets and didn’t analyze them. We recognize our own shortcomings. According to your suggestion, after introducing the SAR data set, then we analyze the data set. The perspectives of our analysis are: the category of data sets, the research on data sets, and the defect of data sets. We selected some representative references to analyze this part. The results of our analysis are as follows (in the last paragraph of Page 13 in subsection 5.1.1):

“According to the above five types of data sets, MSTAR and OpenSARShip are data sets with target type information, and the other three are data sets without target type information. Compared with other public datasets, the MSTAR dataset has a higher resolution and an earlier time to open. Therefore, MSTAR dataset is the most widely used data set in the field of SAR image target detection. The first is sample expansion of MSTAR dataset. Song, Q et al.[43] used generative adversarial networks and adversarial auto-encoders to enhance the MSTAR dataset. In addition, the related research about the MSTAR dataset also includes the improvement of CNN, the research of transfer learning and so on. The improvement of CNN is the main research direction. In the ship target detection data set, SSDD is a kind of data set that is published earlier. Since SSDD belongs to the data set without target category information, the research on this data set is mainly in two directions: ship target detection and target segmentation. Nowadays, most researchers use SSDD to evaluate the proposed model, which can be found from the simulation experiments in some references. In the follow-up study of SSDD, researchers also labeled the target position after rotation, making the labeled information more accurate. But SSDD also have a drawback, that is, the amount of data is too small. It’s prone to overfitting when training the model directly by using SSDD. Therefore, in practice, it is generally used in combination with other data sets to make the model possess better performance. Compared with MATSR and SSDD, SAR-Ship-Dataset and HRSID are published relatively late, so there are few studies on this data set. For SAR-Ship-Dataset, its production team studied this dataset. Reference [44] proposed to take advantage of the RetinaNet for SAR image target detection and used feature pyramid structure to extract multi-scale features. For HRSID, Reference [45] proposed to generate the simulated SAR images for target detection task through sample migration and data migration, which increases the amount of data in the data set and increases the processing complexity. Finally, for OpenSARship, the research focus is target recognition based on semi-supervised learning. In [46], the authors proposed a semi-supervised learning method based on generative adversarial network, which effectively solved the over-fitting problem of complex networks caused by the small number of labeled target samples. The authors used 80%, 60%, 40%, and 20% of the labeled data in the dataset for experiments. Compared with the previous random initialization method, the experimental results show that the recognition accuracy can be increased by 23.58%.”

Please see all the modifications with the highlighted RED font in the revised version.

Thanks,

Ying Zhang

Shanghai Maritime University

2022.11.20

Round 2

Reviewer 1 Report

My All question are answered and I have one more further suggestion to reduce plagiarism by up to 15%. and some minor other suggestion 

1.  please improve the abstract as per the theme of the paper.

2. it is better to cite this related paper about the SAR SLR review paper so cite it in your paper.

Author Response

Authors' Reply Letter for AE and Reviewers

Dear Prof. Jasmine Jia and reviewers,

Thank you for your time and effort put in reviewing our manuscript entitled “A Survey of SAR Image Target Detection Based on Convolutional Neural Networks” (Manuscript ID: remotesensing-2031165). We also thank you for your detailed comments that helped us improve the quality of this paper. In the following, we address your concerns in the order that they were mentioned.

Sincerely,

Manuscript ID: remotesensing-2031165.

Reviewer #1:

My All question are answered and I have one more further suggestion to reduce plagiarism by up to 15%. and some minor other suggestion 

Q(1): I have one more further suggestion to reduce plagiarism by up to 15%.

[Response]: Thank you for your suggestion. In the previous revisions, we don’t pay enough attention to the repetition rate of the article. According to your suggestion, we have reduced the repetition rate of the whole article. We believe that after this modification, the repetition rate of the article meets your requirements.

Q(2): please improve the abstract as per the theme of the paper.

[Response]: Thanks for your opinion. Indeed, as you said, the previous abstract didn’t fit the theme of the article. This is our deficiency. Therefore, according to your suggestion, we rewrite the abstract of the article. We believe that the revised abstract is more consistent with the main content of the paper. The revised abstract is as follows:

Abstract: Synthetic Aperture Radar (SAR) target detection is a significant research direction in the radar information processing. Aiming at the poor robustness and low detection accuracy of traditional detection algorithms, the SAR image target detection based on the Convolutional Neural Network (CNN) is reviewed in this paper. Firstly, the traditional SAR image target detection algorithms are briefly discussed, and their limitations are pointed out. Secondly, the CNN’s network principle, basic structure and development process in the computer vision are introduced. Next, the SAR target detection based on CNN is emphatically analyzed, including some common data sets and image processing methods for SAR target detection. The research status of SAR image target detection based on CNN is summarized and compared in detail with traditional algorithms. Afterwards, the challenges of SAR image target detection are discussed and the future is prospected. Eventually, the whole article is summarized. By summarizing and analyzing prior research work, this paper is helpful for subsequent researchers to rapidly recognize the current development status and identify the connections between various detection algorithms. Beyond that, this paper summarizes the problems and challenges confronted by researchers in the future, and also points out the specific content of the next research, which has certain guiding significance for promoting the progress of SAR image target detection.

Q(3): it is better to cite this related paper about the SAR SLR review paper so cite it in your paper.

[Response]: Thanks for the reminder. In our last revision, we added the Chapter 2 (Research Methodology) into the article. But we have neglected the corresponding references. This is really our mistake. Therefore, according to your suggestion, we have added this reference, which is the reference in the article [7].

Thank you again for your suggestions for this article.

Please see all the modifications with the highlighted RED font in the revised version.

Thanks,

Ying Zhang

Shanghai Maritime University

2022.12.01

Reviewer 2 Report

Dear authors,

Thanks for your re-submission. The comments are as follows,

1. Please add 'Author Contributions', 'Conflicts of Interest', or other part in the end of this manuscript.

2. Please double check all references format. like, ref. 9, 20, 27, 32, and so on.

Author Response

Authors' Reply Letter for AE and Reviewers

Dear Prof. Jasmine Jia and reviewers,

Thank you for your time and effort put in reviewing our manuscript entitled “A Survey of SAR Image Target Detection Based on Convolutional Neural Networks” (Manuscript ID: remotesensing-2031165). We also thank you for your detailed comments that helped us improve the quality of this paper. In the following, we address your concerns in the order that they were mentioned.

Sincerely,

Manuscript ID: remotesensing-2031165.

Reviewer #2:

Dear authors,

Thanks for your re-submission. The comments are as follows,

Q(1): Please add 'Author Contributions', 'Conflicts of Interest', or other part in the end of this manuscript.

[Response]: Thank you very much for your suggestions for this article. According to your suggestion, we have added Author Contributions, Acknowledgments, Conflicts of Interest at the end of the article. This can make the article more complete. The contents added are as follows:

Author Contributions Y.Z. put forward the general framework of the article and provided the writing ideas, conceived and supervised the research and experiments, contributed as the lead author of the article. At the same time, for the improvement and revision of the article, Y.Z. also made many constructive comments. Y.H. consulted the references and completed the writing and revision of this article. All authors have read and agreed to the published version of the manuscript.

Funding:  This work was supported by the National Natural Science Foundation of China (no. 61673259); supported by Shanghai "Science and Technology Innovation Action Plan" Hong Kong, Macao and Taiwan Science and Technology Cooperation Project (no.21510760600); and also supported by Capacity Building Project of Local Colleges and Universities of Shanghai (no. 21010501900).

Acknowledgments: The authors would like to thank the College of Information Engineering and Institute of Logistics Science and Engineering of Shanghai Maritime University for their support. The author would also like to thank the anonymous reviewers for their helpful suggestions and comments to improve the article.

Institutional Review Board Statement: Not applicable.

Informed Consent Statement: Not applicable.

Conflicts of Interest: The authors declare no conflict of interest.

Q(2): Please double check all references format. like, ref. 9, 20, 27, 32, and so on.

[Response]: Thanks for your suggestion. We are very sorry to cause you some trouble due to our negligence. It is true that some references are not normal. We have modified some of the references. We believe that the revised Reference part basically meets your requirements, please see the revised version of the manuscript.

Finally, thank you again for your many constructive comments for this article.

Please see all the modifications with the highlighted RED font in the revised version.

Thanks,

Ying Zhang

Shanghai Maritime University

2022.12.01

Reviewer 3 Report

Thank you for your detailed revisions which do indeed address the original concerns.

Author Response

Authors' Reply Letter for AE and Reviewers

Dear Prof. Jasmine Jia and reviewers,

Thank you for your time and effort put in reviewing our manuscript entitled “A Survey of SAR Image Target Detection Based on Convolutional Neural Networks” (Manuscript ID: remotesensing-2031165). We also thank you for your detailed comments that helped us improve the quality of this paper. In the following, we address your concerns in the order that they were mentioned.

Sincerely,

Manuscript ID: remotesensing-2031165.

Reviewer #3:

Thank you for your detailed revisions which do indeed address the original concerns.

[Response]: Sincerely thanks for your affirmation of our work, also thank you for your comments and suggestions for this article.

Thanks,

Ying Zhang

Shanghai Maritime University

2022.12.01
